# MaskCLR: Multi-Level Contrastive Learning for Robust Skeletal Action Recognition

## Abstract

Current transformer-based skeletal action recognition models focus on a limited set of joints and low-level motion patterns to predict action classes. This results in significant performance degradation under small skeleton perturbations or changing the pose estimator between training and testing. In this work, we introduce **MaskCLR**, a new **Mask**ed **C**ontrastive **L**earning approach for **R**obust skeletal action recognition. We propose a Targeted Masking (TM) strategy to occlude the most important joints and encourage the model to explore a larger set of discriminative joints. Furthermore, we propose a Multi-Level Contrastive Learning (MLCL) paradigm to enforce feature embeddings of standard and occluded skeletons to be class-discriminative, *i.e,* more compact within each class and more dispersed across different classes. Our approach helps the model capture the high-level action semantics instead of low-level joint variations, and can be seamlessly incorporated into transformer-based models. Without loss of generality, we apply our method on $\mathcal{S}$patial-$\mathcal{T}$emporal Multi-Head Self-Attention encoder ($\mathcal{ST}$-MHSA), and we perform extensive experiments on NTU60, NTU120, and Kinetics400 benchmarks. MaskCLR consistently outperforms previous state-of-the-art methods on standard and perturbed skeletons from different pose estimators, showing improved accuracy, generalization, and robustness to skeleton perturbations. We make our implementation anonymously available at anonymous.4open.science/r/MaskCLR-A503.

## 1 Introduction

A skeleton is a representation of the human body structure that typically consists of a set of keypoints or joints, each associated with a specific body part. Compared to RGB-based action recognition, which focuses on extracting feature representations from RGB frames (Carreira & Zisserman, 2017; Tran et al., 2015; Wang et al., 2016) and/or optical flow (Simonyan & Zisserman, 2014), skeleton-based approaches (Weinzaepfel & Rogez, 2021; Yan et al., 2018) rely on skeleton data. Skeleton sequences exclude contextual nuisances such as lighting conditions and background changes, and are therefore more compact, easier to store, and more computationally efficient. The skeleton data can be either in 2D or 3D and it can be extracted from RGB images using various pose estimators or directly captured by sensor devices such as Kinect (Liu et al., 2019; Shahroudy et al., 2016).

With the introduction of transformers (Vaswani et al., 2017), $\mathcal{S}$patial and $\mathcal{T}$emporal Multi-Head Self Attention ($\mathcal{ST}$-MHSA) blocks (Zhu et al., 2023) have been proposed to extract the spatiotemporal skeleton information for action recognition. Generally, $\mathcal{ST}$-MHSA blocks give higher weights to the most important joints/input regions that characterize every action to distinguish between different classes. For example, the hand joints in the action "throw" receive the highest weights while the rest of joints remain relatively unactivated. Motivated by this observation, we ask: *Is it possible to exploit the information carried in the unactivated joints to aid in action classification?*

To answer this question, in Figure 1 (top row), we visualize the activated joints of different samples according to their attention weights. The attention scores are learned by State-Of-The-Art (SOTA) transformer-based MotionBERT (Zhu et al., 2023). From the visualization, we observe that the model focuses on a limited set of discriminative joints to recognize the actions. Because of this, we argue that the model (1) misses action semantics (2) misclassifies the action if such joints are slightly perturbed and (3) ignores other joints which might be informative in action classification (*e.g,* in

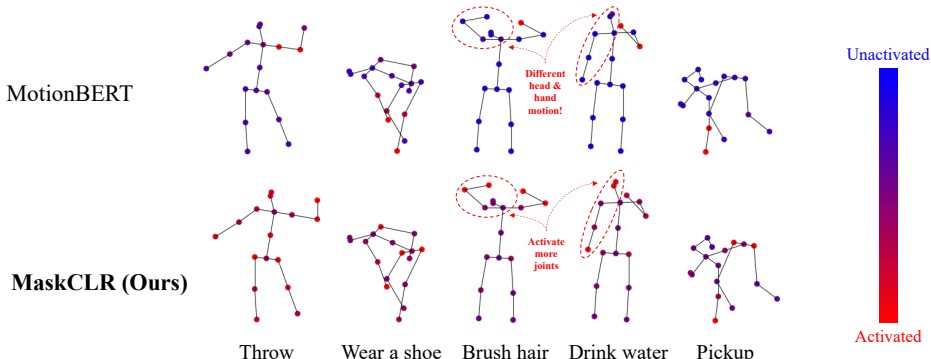

Figure 1: **Visualization of joint activations of MotionBERT (Zhu et al., 2023) and MaskCLR.** The activated joints are more reddish while the unactivated ones are more blueish. MaskCLR uses a bigger set of discriminative joints to recognize actions.

"brush hair" and "drink water," the unactivated joints carry useful information about head and hand motion, which is significantly different (see Figure 1)).

Consequently, these methods fall short in the following aspects, summarized in table 1:

(1) *Robustness against skeleton perturbations*: the recognition accuracy of existing methods is substantially affected by action-preserving levels of perturbations. For example, a small shift in joint coordinates often leads to a completely different classification prediction.

(2) *Generalization to pose source*: Changing the pose estimator used to extract the skeletons between training and testing results in a considerable drop in accuracy. This shows that such methods only model the distribution of the predicted joints from the specific pose estimator used for training data extraction, but fail to handle any distribution shift from using a different pose estimator at test time.

Table 1: **Failure cases of current methods on different skeleton sequences.**

| Pose Estimator | Raw Skeletons | Perturbed Skeletons |
|---|:---:|:---:|
| Same in training & testing | ✓ | ✗ |
| Changed between training & testing | ✗ | ✗ |

In this paper, we introduce **MaskCLR**, a novel masked contrastive learning framework that improves the robustness, accuracy, and generalization of transformer-based methods. First, instead of using only a few joints to recognize actions, we propose a **Targeted Masking (TM)** strategy to occlude the most activated joints and re-feed the resulting skeletons to the model. This strategy aims at *forcing* the model to explore a bigger set of informative joints out of the unactivated ones. Further, we propose a **Multi-Level Contrastive Learning (MLCL)** approach, which consists of two flavours of contrastive losses: sample- and class-level contrastive losses. At the sample level, we maximize the similarity between the embeddings of standard and masked skeleton sequences in the feature space. At the class level, we take advantage of the cross-sequence global context by contrasting the class-averaged features of standard and masked skeleton sequences.

MaskCLR directly addresses the aforementioned limitations of existing methods. By utilizing the previously unactivated, yet informative joints, our method helps the model learn the holistic motion patterns of multiple joints (see Figure 1 (bottom row)). Instead of relying on the motion of a few discriminative joints that is easily damaged by a small noise, using more informative joints mitigates the effect of action-preserving levels of noise. Our technique, therefore, boosts the overall model accuracy and robustness against common skeleton perturbations, such as joint occlusion and frame masking. Moreover, our MLCL paradigm leverages the inherent class-wise semantic information in forming a class-discriminative embedding space. Since raw and perturbed skeletons extracted from different pose estimators reflect similar action semantics, our framework substantially improves robustness against noisy skeletons and generalization to the type of pose estimator. To the best

of our knowledge, MaskCLR is the first approach that improves robustness and generalization of transformer-based skeletal action recognition. Notably, MaskCLR only requires a small extra amount of training computation, but does not change the model size or inference time.

To summarize, our key contributions in MaskCLR are threefold:

- First, we propose a TM strategy aimed at finding and masking the most activated joints from the skeleton sequence. We pass the resulting masked sequence through the model to learn from the unactivated, but informative joints. Our objective is to recognize the combined joint motion patterns instead of focusing on a small set of joints.

- Next, we introduce an MLCL paradigm to leverage the rich semantic information shared in skeleton sequences of the same class. Our approach results in a better clustered feature space which boosts the overall model performance.

- Finally, we demonstrate through extensive experimental results the superiority of our approach over existing methods on three popular action recognition benchmarks (NTU60 Shahroudy et al. (2016), NTU120 Liu et al. (2019), and Kinetics400 Kay et al. (2017)). MaskCLR consistently outperforms previous methods under heavy skeleton perturbations and pose estimator changes.

## 2 RELATED WORKS

### 2.1 SKELETON-BASED ACTION RECOGNITION

The main objective behind skeleton-based action recognition is to classify a sequence of human keypoints into a set of action categories. Convolutional Neural Networks (CNNs) (Chéron et al., 2015; Liu et al., 2017b) and Recurrent Neural Networks (RNNs) (Du et al., 2015; Liu et al., 2017a) were among the earliest adopted deep-learning methods to model the spatiotemporal correlations in the skeletons but the performance was suboptimal because the topological structure of the skeletons was not well explored. Significant performance gains were obtained by employing Graph Neural Networks (GCNs) as a feature extractor on heuristically designed fixed skeleton graphs, which was first introduced in ST-GCN (Yan et al., 2018). Since then, numerous methods have emerged to improve the accuracy and robustness of GCNs, including the usage of spatiotemporal graphs (Liu et al., 2020), channel-decoupled graphs (Chen et al., 2021a; Cheng et al., 2020a), multi-scale graph convolution (Chen et al., 2021b), and adaptive graphs (Chi et al., 2022; Shi et al., 2020). More recently, PoseConv3D (Duan et al., 2022b) re-introduced 3D-CNNs for action recognition by projecting skeletons into stacked 3D Heatmaps. Compared to GCNs, PoseConv3D obtained significant improvements in robustness but marginal improvements in accuracy. Transformers have also been adopted for action recognition, most recently in MotionBERT (Zhu et al., 2023) which performs 2D-to-3D pose lifting to learn motion representations and Motion-Transformer (Cheng et al., 2021) which employs self-supervised pre-training on human actions to learn temporal dependencies.

However, these methods (1) lack robustness against perturbed skeletons which are fairly common in real world applications, and (2) cannot handle the distribution shift in poses from a different pose estimator at test time. Additionally, transformer-based methods (3) give much higher weights to a small set of joints without leveraging the information carried by the other joints, and (4) focus only on learning local graph representations but neglect the rich semantic information shared between skeleton sequences of the same classes. In contrast, we propose to take advantage of the cross-sequence correlations in learning skeleton representations. At the same time, we suggest to rely on more joints in differentiating action classes to encode the local context within each skeleton sequence. In this way, we learn a more robust model that better handles skeleton perturbations and distribution shifts from changing the pose estimator between training and testing.

### 2.2 CONTRASTIVE LEARNING

The core idea behind contrastive learning is to pull together representations of similar inputs (positive pairs) while pushing apart that of dissimilar ones (negative pairs) in the feature space. It has been shown to contribute for substantial performance gains, especially in self-supervised representation learning (Wang et al., 2021; Chen et al., 2020; He et al., 2020). Positive pairs are conventionally

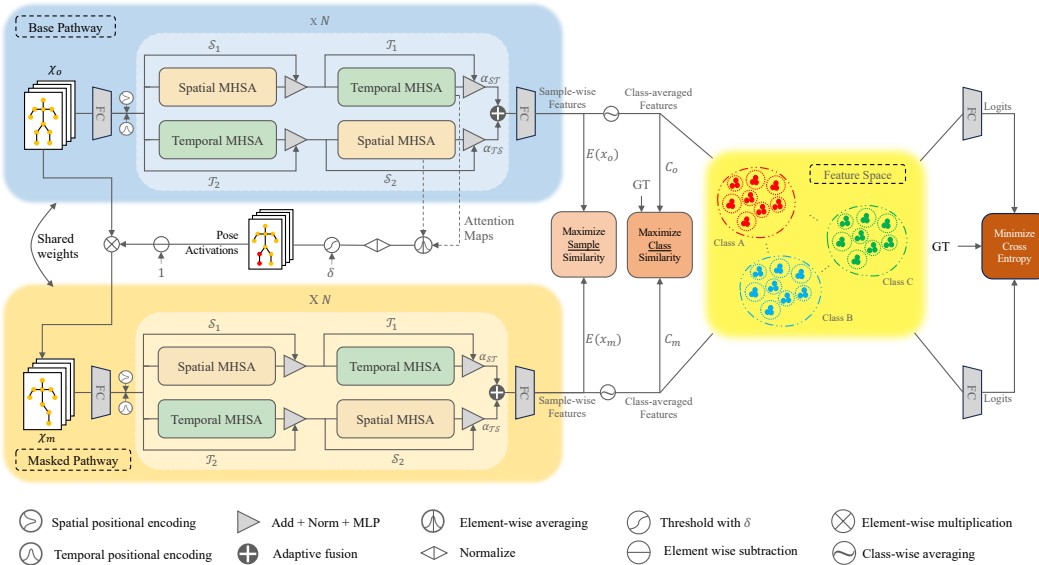

Figure 2: **Overview of MaskCLR.** Our approach consists of two (*base* and *masked*) pathways that share the same weights. The base pathway takes standard input skeletons while the masked pathway receives only the unactivated joints from the base pathway. Both pathways consist of cascaded spatial and temporal MHSA blocks to extract the spatial and temporal information respectively. Initially, the base pathway is trained alone using cross-entropy loss. The masked pathway, subsequently, comes into play to encourage the model to explore more discriminative joints. Using sample contrastive loss, we maximize the agreement of feature representations from the two pathways for the same skeleton sequence and vice versa. Additionally, to exploit the high semantic consistency between same-class skeleton sequences, we maximize the similarity between the class-wise average representations from the two pathways using class contrastive loss. Ultimately, the two contrastive losses contribute to the formation of a disentangled feature space, effectively improving the accuracy, robustness, and generalization of the model.

obtained by augmenting the standard input into two different views, while negative ones are obtained either through random sampling or hard mining techniques (Kalantidis et al., 2020; Robinson et al., 2020; Khosla et al., 2020). In skeleton-based action recognition, the aforementioned frameworks have been adopted in the pre-training stage. In CrossCLR (Li et al., 2021), the positive pairs are sampled in the data space by cross-modal knowledge. AimCLR (Guo et al., 2022) uses extreme augmentations to boost the effect of contrastive learning. More recently, ActCLR (Lin et al., 2023) uses the average motion across all sequences in the dataset as a static anchor for contrastive learning.

Our method differs from these approaches as follows: (1) The previous methods sample positive pairs by using fixed sample-wide augmentations that are invariant to the internal semantics of the action. In contrast, we employ a new *adaptive masking strategy* for sampling the positive pairs by occluding the most activated joints, which vary based on the input sample and action. (2) The previous methods employ contrastive learning at the sample level only. Instead, we contrast the semantic-level *class* representations, thus exploiting the context from the complementary individual and class aggregations. (3) While the previous methods employ contrastive learning in the pre-training stage, MaskCLR is incorporated in the *fully-supervised* setting, thus requiring no extra pre-training cost.

## 3 METHOD

In this section, we introduce MaskCLR, our novel approach to enhance the accuracy, robustness, and generalization of transformer-based skeletal action recognition methods. MaskCLR consists of a targeted masking strategy (Sec 3.1) combined with a multi-level contrastive learning approach (Sec 3.2 & Sec 3.3). As shown in Figure 2, our approach consists of two pathways: a base pathway, which receives standard skeletons as input, and a masked one, which receives only the unactivated joints from the base pathway.

## 3.1 TARGETED MASKING STRATEGY

We leverage the $\mathcal{S}$patial $\mathcal{T}$emporal Multi-Head Self-Attention ($\mathcal{ST}$-MHSA) backbone of transformer-based models (Zhu et al., 2023; Plizzari et al., 2021) to compute the joint-wise attention weights, which reflect the joints' importance over the spatiotemporal dimensions. First, an input 2D skeleton sequence $\mathbf{x}$ of $T$ frames and $J$ joints is fed to a Fully Connected (FC) network to get the high-dimensional feature $\mathbf{F} \in \mathbb{R}^{T \times J \times C_f}$ of $C_f$ channels. We then apply the $\mathcal{ST}$-MHSA encoder for $N$ times on $\mathbf{F}$ before passing the output to an FC network to get the sequence $\mathbf{x}$ embedding $\mathbf{E}(\mathbf{x}) \in \mathbb{R}^{T \times J \times C_r}$ of $C_r$ channels. Each MHSA block is composed of $h$ heads defined as

$$head_b^i = softmax \left( \frac{\mathbf{Q}_\mathbf{b}^i (\mathbf{K}_\mathbf{b}^i)^T}{\sqrt{d_K}} \right) \mathbf{V}_\mathbf{b}^i, \tag{1}$$

where $i \in 1, ..., h$ denotes the attention head, and $b \in \mathcal{S}, \mathcal{T}$ denotes spatial and temporal blocks respectively. Self-attention is utilized to calculate the query $\mathbf{Q}_\mathbf{b}$, key $\mathbf{K}_\mathbf{b}$, and value $\mathbf{V}_\mathbf{b}$ from input features $\mathbf{F}_\mathbf{b}$, where $\mathbf{F}_\mathcal{S} \in \mathbb{R}^{J \times C_r}$, $\mathbf{F}_\mathcal{T} \in \mathbb{R}^{T \times C_r}$, and $d_K$ is the dimension of $\mathbf{K}_\mathbf{b}$. In our approach, the attention scores are computed by averaging the attention maps $\mathbf{A}$ across the self-attention heads.

$$\mathbf{A}_b = \frac{1}{h} \sum_{i=1}^{h} softmax \left( \frac{\mathbf{Q}_\mathbf{b}^i (\mathbf{K}_\mathbf{b}^i)^T}{\sqrt{d_K}} \right). \tag{2}$$

Only the last $\mathcal{S}$ and $\mathcal{T}$ blocks are used to compute the most activated/important joints for targeted masking since they inherit the information learned from the previous layers. In the Appendix Sec A.2, we study the effect of using the attention filters from other layers. The attention scores $\mathbf{A}_b$ are averaged across the final two $\mathcal{ST}$ blocks $\mathbf{A} = 1/2 * (\mathbf{A}_\mathcal{S} + \mathbf{A}_\mathcal{T})$. Next, a predefined threshold $\delta \in [0, 1]$ is utilized to determine the joints activated by the base pathway $\mathbf{P}$:

$$\mathbf{P} = \varepsilon \left( \frac{\mathbf{A}}{max(\mathbf{A})} - \delta \right), \tag{3}$$

where $\varepsilon(.)$ is the element-wise Heaviside step function, and $max(.)$ is the maximum function used to normalize the attention scores. Hence, $\mathbf{P} \in \mathbb{R}^{T \times J}$ is a binary map indicating the activated joints across time. Visualization of the activated joints (before thresholding) is provided in Figure 1. We expand the set of informative joints by masking the most activated joints from the base pathway and feeding the unactivated ones to the masked pathway (see Figure 2), thus encouraging the model to explore more discriminative joints (Figure 1 bottom). The calculation of masked skeletons $\mathbf{x}_m$ from the original skeletons $\mathbf{x}_o$ is formalized as

$$\mathbf{x}_m = \mathbf{x}_o \otimes (1 - \mathbf{P}). \tag{4}$$

## 3.2 SAMPLE CONTRASTIVE LOSS

Having computed the more challenging masked skeletons $\mathbf{x}_m$, our target is to achieve a model that can learn from the unactivated joints. We observe that such joints carry information about the body pose which could inform the action prediction (see Figure 1). To that end, we adopt sample contrastive loss to maximize the similarity between the embeddings of standard $\mathbf{E}(\mathbf{x}_o^i)$ and masked $\mathbf{E}(\mathbf{x}_m^i)$ skeletons which correspond to the same skeleton sequence $i$. More specifically, for a batch of size $B$, the positive pairs are $\mathbf{E}(\mathbf{x}_o^i)$ and $\mathbf{E}(\mathbf{x}_m^i)$ while the negative ones are the rest of $B - 1$ pairs, $\mathbf{E}(\mathbf{x}_o^i)$ and $\mathbf{E}(\mathbf{x}_z^k)$, $k \neq i, z \in \{o, m\}$. Since skeletons extracted from different videos, forming the negative pairs, have different content, the similarity of their representations should be minimized in the latent space. We achieve this objective by applying the sample contrastive loss $\mathcal{L}_{sc}$.

$$\mathcal{L}_{sc}(\mathbf{x}_o^i, \mathbf{x}_m^i) = -\log \left[ \frac{s(\mathbf{E}(\mathbf{x}_o^i), \mathbf{E}(\mathbf{x}_m^i))}{s(\mathbf{E}(\mathbf{x}_o^i), \mathbf{E}(\mathbf{x}_m^i)) + \sum_{k=1}^{B} \mathbb{1}_{k \neq i} s(\mathbf{E}(\mathbf{x}_o^i), \mathbf{E}(\mathbf{x}_z^k))} \right], \tag{5}$$

where $\mathbb{1}$ is an indicator function that evaluates to 1 for skeletons corresponding to a different sample $k \neq i$, $s$ is the exponential of the cosine similarity $s(u, v) = \exp\left[\frac{u^T v}{\|u\|_2 \|v\|_2} / \tau\right]$, and $\tau$ is the temperature hyperparameter.

## 3.3 CLASS CONTRASTIVE LOSS

Used alone, the sample contrastive loss function encourages representations of different skeleton sequences to be pushed apart even if they belong to the same action class. This, in turn, could result in overlooking the high-level semantics that characterize every action. Therefore, we need to maximize the similarity between skeleton sequences sharing the same class. At the same time, skeleton sequences of different classes need to be pushed apart in the feature space, especially in the case of semantically similar classes (*e.g,* reading and writing), which are often confused together in existing models. We introduce class contrastive loss as a complementary loss function to achieve both objectives simultaneously. Similar to sample contrastive loss, class contrastive loss encourages distinct representations but at the class level. Class representation $\mathbf{C}_z^l$ is defined as the average embeddings of all skeleton sequences sharing the same label $l$ across every batch. $\mathbf{C}_z^l$ formalized as:

$$\mathbf{C}_z^l = \frac{\sum_{i=1}^{B} \mathbb{1}_{y^i = l} \mathbf{E}(\mathbf{x}_z^i)}{G^l},\tag{6}$$

where $z \in \{o, m\}$, $y^i \in Y$ is ground truth label of sample $i$, $Y$ is the set of dataset classes, and $\mathbb{1}$ is an indicator function which evaluates to 1 for skeleton sequences with label $l$. $G^l$ is the number of skeleton sequences sharing the label $l$ across the batch.

All pairs $(\mathbf{C}_o^l, \mathbf{C}_m^l)$ constitute the positive pairs, while $(\mathbf{C}_o^l, \mathbf{C}_z^d)$ with $d \in Y \setminus l$ constitute the negative ones. The class contrastive objective is to capture the high-level action semantics in skeleton sequences sharing the same class. Hence, the class contrastive loss $\mathcal{L}_{cc}$ is:

$$\mathcal{L}_{cc}(\mathbf{C}_o^l, \mathbf{C}_m^l) = -\log\left[\frac{s(\mathbf{C}_o^l, \mathbf{C}_m^l)}{s(\mathbf{C}_o^l, \mathbf{C}_m^l) + \sum_{k=1}^{|Y|} \mathbb{1}_{k \neq l} s(\mathbf{C}_o^l, \mathbf{C}_z^k)}\right].\tag{7}$$

**Overall loss.** Finally, the overall loss function used to train our model is,

$$\mathcal{L} = \mathcal{L}_{ce} + \alpha \mathcal{L}_{sc} + \beta \mathcal{L}_{cc},\tag{8}$$

where $\mathcal{L}_{ce}$ is the average cross entropy loss from the two pathways, and $\alpha$ and $\beta$ are the weights assigned to sample and class contrastive losses respectively.

# 4 EXPERIMENTS

## 4.1 DATASETS

We use the **NTU RGB+D** (Shahroudy et al., 2016; Liu et al., 2019) and **Kinetics400** (Kay et al., 2017) datasets in our experiments. To obtain 2D poses, we employ three pose estimators (pre-trained on MS COCO (Lin et al., 2014)) of different AP scores: **ViTPose** (SOTA) (Xu et al., 2022) (High Quality, HQ), **HRNet** (Sun et al., 2019) (Medium Quality, MQ), and **OpenPifPaf** (Kreiss et al., 2021) (Low Quality, LQ). We apply the same post-processing across the three versions (outlier removal, pose tracking, etc). Further details about the datasets and pose estimators are provided in the Appendix Sec A.1. We report the Top-1 accuracy for all datasets.

## 4.2 IMPLEMENTATION DETAILS

We implement our approach with PyTorch (Paszke et al., 2019) on top of spatial and temporal transformer encoder, borrowed from (Zhu et al., 2023). We set the depth $N = 5$, $C_f = 512$,

$C_r = 512$, $h = 8$, and fix temporal sampling at $T = 243$. Different temporal lengths could be handled at test time due to the flexibility of the transformer backbone. For contrastive losses, we set $\alpha = 9$, $\beta = 1$, $\delta = 0.2$ and $\tau = 0.7$. The classification head is an MLP with hidden dimension = 2048, drop out rate $p = 0.5$, BatchNorm, and ReLU activation. We train our model for 600 epochs, where we first use only $\mathcal{L}_{ce}$ to train the base pathway for 300 epochs. For the next 300 epochs, we add the masked pathway and train the model with the combined loss (Eq. 8). We train with backbone learning rate 0.0001, MLP learning rate 0.001, and batch size 16 using AdamW (Loshchilov & Hutter, 2017) optimizer. We conduct our experiments with two NVIDIA RTX 3090 GPUs.

## 4.3 COMPARISON WITH STATE-OF-THE-ART METHODS

**Accuracy on standard skeletons.** In Table 2, we compare the accuracy of MaskCLR to existing methods under the fully-supervised setting. MaskCLR outperforms previous SOTA methods on **3 out of 5** benchmarks, and outperforms baseline MotionBERT (Zhu et al., 2023) on **all** benchmarks. For NTU120-XSet, MaskCLR is only 0.8 short of PoseConv3D, yet improves the accuracy of MotionBERT by **2.4** percentage points. For Kinetics, MaskCLR surpasses MotionBERT by a margin of **5.8** percentage points. This shows that MaskCLR improves the accuracy on standard skeletons at no pre-training cost and without increasing the model size.

Table 2: **MaskCLR outperforms or closely competes with previous SOTA.** Numbers in green reflect improvement over baseline MotionBERT.

| Method | NTU60-XSub | NTU60-XView | NTU120-XSub | NTU120-XSet | Kinetics400 |
|---|---|---|---|---|---|
| ST-GCN (Yan et al., 2018) | 81.5 | 88.3 | 70.7 | 73.2 | 30.7 |
| AS-GCN (Li et al., 2019) | 86.8 | 94.2 | 78.3 | 79.8 | 34.8 |
| DGNN (Shi et al., 2019a) | 89.9 | 96.1 | - | - | 36.9 |
| AGCN (Shi et al., 2019b) | 88.5 | 95.1 | - | - | 36.1 |
| AAGCN (Shi et al., 2020) | 89.7 | 97.1 | 80.2 | 86.3 | - |
| Shift-GCN (Cheng et al., 2020b) | 90.7 | 96.5 | 85.9 | 87.6 | - |
| MS-G3D (Liu et al., 2020) | 92.2 | 96.6 | 87.2 | 89.0 | 45.1 |
| FGCN (Yang et al., 2021) | 90.2 | 96.3 | 85.4 | 87.4 | - |
| CTR-GCN (Chen et al., 2021a) | 90.6 | 96.9 | 82.2 | 84.5 | - |
| AimCLR (Guo et al., 2022) | 89.2 | 83.0 | 76.1 | 77.2 | - |
| ST-GCN++ (Duan et al., 2022a) | 89.3 | 95.6 | 83.2 | 85.6 | - |
| InfoGCN (Chi et al., 2022) | 93.0 | 97.1 | 85.1 | 86.3 | - |
| PoseConv3D (Duan et al., 2022b) | 93.7 | 96.6 | 86.0 | **89.6** | **46.0** |
| ActCLR (Lin et al., 2023) | 91.2 | 85.8 | 80.9 | 79.4 | - |
| FR-GCN (Zhou et al., 2023) | 90.3 | 95.3 | 85.5 | 88.1 | - |
| MotionBERT (Zhu et al., 2023) | 92.8 | 97.1 | 84.8 | 86.4 | 38.8 |
| **MaskCLR (Ours)** | **93.9** (↑1.1) | **97.3** (↑ 0.2) | **87.4** (↑2.6) | 88.8 (↑2.4) | 44.7 (↑5.8) |

**Generalization to pose source.** To demonstrate the generalization of our method to the type of pose estimator, we evaluate our model on the skeletons extracted by different pose estimators of different quality levels. For a fair comparison, we train all models on MQ skeletons and we evaluate on LQ and HQ ones (see Table 3). MaskCLR consistently outperforms previous methods on **all** benchmarks under HQ and LQ poses. Compared to MotionBERT, we improve generalization to skeleton source by up to **26.6** percentage points for NTU120-XSub VitPose Skeletons.

Table 3: **Top-1 accuracy when changing the pose estimator between training and testing.** Numbers in green reflect improvement over baseline MotionBERT. Right arrows indicate train → test.

| Method | NTU60 | | NTU120 | | NTU60 | | NTU120 | |
|---|---|---|---|---|---|---|---|---|
| | XSub | XView | XSub | XSet | XSub | XView | XSub | XSet |
| Pose Source | HRNet (MQ) → OpenPifPaf (LQ) | | | | HRNet (MQ) → ViTPose (HQ) | | | |
| ST-GCN++ | 65.3 | 72.4 | 68.8 | 71.2 | 73.2 | 83.0 | 66.4 | 69.6 |
| MS-G3D | 51.7 | 57.3 | 54.9 | 55.8 | 82.5 | 91.0 | 72.1 | 74.5 |
| AAGCN | 51.4 | 60.1 | 51.5 | 62.7 | 79.6 | 90.2 | 66.6 | 72.0 |
| CTR-GCN | 50.4 | 58.2 | 57.6 | 58.2 | 77.7 | 84.4 | 65.5 | 67.7 |
| PoseConv3D | 83.2 | 87.3 | 80.2 | 82.9 | 73.7 | 79.8 | 65.2 | 70.1 |
| MotionBERT | 90.4 | 91.9 | 73.5 | 77.6 | 71.0 | 85.8 | 51.7 | 63.9 |
| **MaskCLR** | **93.4** (↑ 3.0) | **97.2** (↑ 5.3) | **87.1** (↑ 13.6) | **86.5** (↑ 8.9) | **91.5** (↑ 20.5) | **96.3** (↑ 10.5) | **78.3** (↑ 26.6) | **79.8** (↑ 15.9) |

**Robustness against skeleton perturbations.** We compare the robustness of our method to existing methods under Gaussian noise, part occlusion, and joint occlusion. For noisy skeletons, we introduce Gaussian noise $X \sim \mathcal{N}(0, \sigma^2)$ to all joints across the spatiotemporal and spatial only dimensions. In

Appendix Figure 7, we provide visualizations of noisy skeletons. As shown in Figure 3, MaskCLR is superior to existing methods under noisy skeletons. Additionally, MaskCLR surpasses MotionBERT-R, which is trained with the same data augmentations and 15% random masking, by **14.9** points at spatiotemporal noise $\sigma = 0.01$. In part occlusion, we separately zero out five body parts {head, left_arm, right_arm, left_leg, right_leg}. In joint occlusion, we randomly mask 15%, 30%, 45%, and 60% of joints. As shown in Figure 4, MaskCLR substantially outperforms existing methods under part and joint occlusion. We provide more experimental results in Appendix Sec A.4 and A.5.

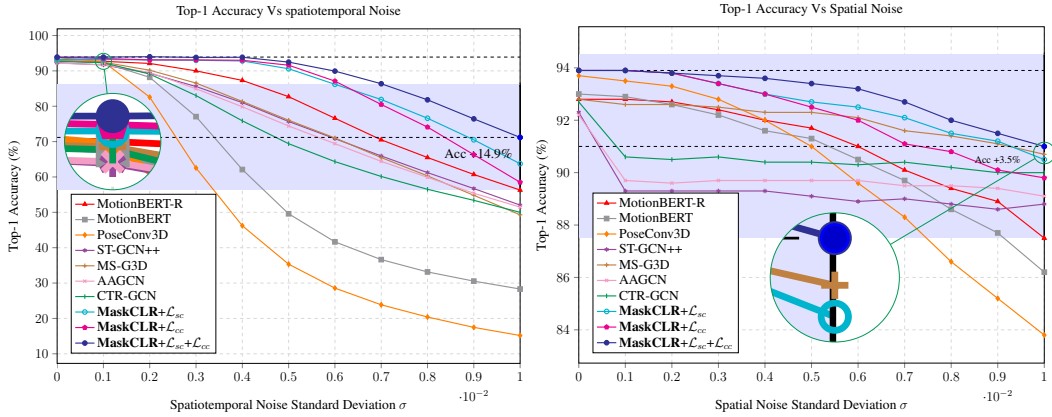

Figure 3: **Accuracy under spatiotemporal noise (left) and spatial only noise (right)**

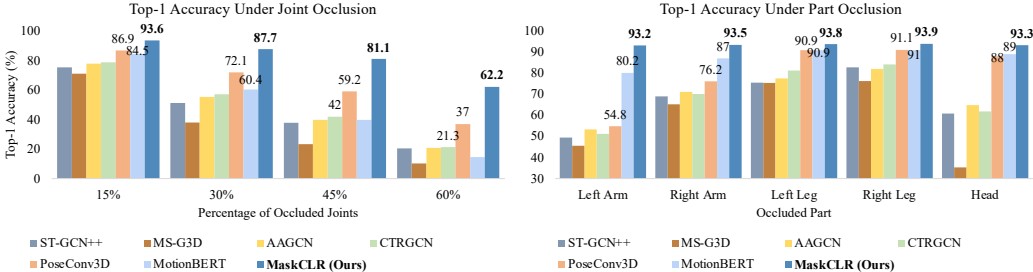

Figure 4: **Top-1 accuracy under joint and part occlusion.** Accuracy indicated for the top 3 methods.

**Robustness against perturbed skeletons from different pose estimators.** We perturb all LQ, MQ, and HQ skeletons with 50% frame masking and spatiotemporal Gaussian noise at $\sigma = 0.002$. While the performances of PoseConv3D (Duan et al., 2022b) and MotionBERT significantly degrades, MaskCLR shows the smallest drop in accuracy (see Table 4). This shows the superiority of our approach in generalization and robustness under perturbed skeletons from different pose estimators.

Table 4: **Drop in top-1 accuracy under skeleton perturbations.** Numbers in green and red reflect improvement and decline compared to baseline MotionBERT, respectively.

| Pose Estimator | Method | NTU60 XSub | NTU60 XView | NTU120 XSub | NTU120 XSet | NTU60 XSub | NTU60 XView | NTU120 XSub | NTU120 XSet |
|---|---|---|---|---|---|---|---|---|---|
| Perturbations | | Gaussian Noise ($\sigma = 0.002$) | | | | 50% Frame Masking | | | |
| HRNet | PoseConv3D | 12.0 | 16.6 | 12.9 | 13.8 | 1.9 | 3.1 | 6.2 | 8.4 |
| | MotionBERT | 4.7 | 8.4 | 3.3 | **0.1** | 1.7 | 1.6 | 1.2 | 4.4 |
| | **MaskCLR** | **0.1** (↑ 4.6) | **0.2** (↑ 8.2) | **0.2** (↑ 3.1) | 1.1 (↓ 1.0) | **1.6** (↑ 0.1) | **1.0** (↑ 0.6) | **0.8** (↑ 0.4) | **4.3** (↑ 0.1) |
| OpenPifPaf | PoseConv3D | 15.3 | 19.9 | 16.6 | 16.7 | 4.0 | 5.0 | 7.9 | 10.0 |
| | MotionBERT | 8.3 | 6.4 | 2.4 | 7.0 | 2.0 | 2.6 | 1.2 | 4.7 |
| | **MaskCLR** | **0.1** (↑ 8.2) | **0.2** (↑ 6.2) | **0.1** (↑ 2.3) | **0.0** (↑ 7.0) | **1.6** (↑ 0.4) | **1.0** (↑ 1.6) | **1.1** (↑ 1.1) | **2.4** (↑ 2.3) |
| ViTPose | PoseConv3D | 9.2 | 13.7 | 2.8 | 10.9 | 2.2 | 2.7 | 9.0 | 7.9 |
| | MotionBERT | 2.9 | 9.4 | 1.9 | 5.0 | **0.9** | 10.2 | 3.1 | 2.6 |
| | **MaskCLR** | **0.1** (↑ 2.8) | **0.4** (↑ 9.0) | **0.7** (↑ 0.2) | **0.1** (↑ 4.9) | 1.1 (↓ 0.2) | **1.1** (↑ 9.1) | **2.3** (↑ 0.8) | **2.1** (↑ 0.5) |

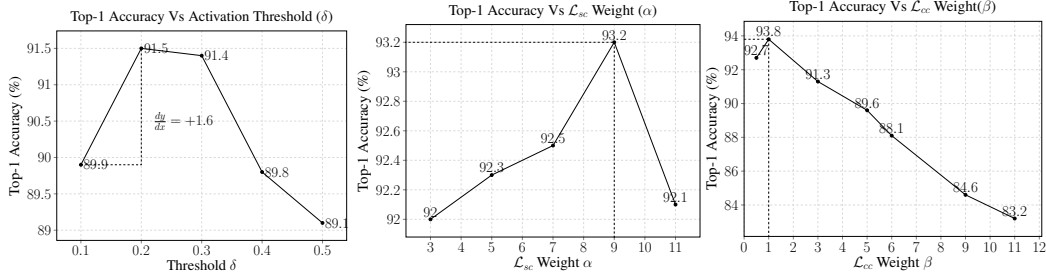

Figure 5: **Effect of hyperparameters on NTU60-XSub.**

## 4.4 ABLATION STUDIES

Next, we perform ablation experiments on NTU60-XSub dataset to better understand the effect of the different components and hyperparameters in our framework. Further ablations are provided in Appendix Sec A.2.

**Random Masking (RM) vs Targeted Masking (TM).** We investigate the effectiveness of our targeted masking strategy against random masking, which is commonly used in previous work (Zhu et al., 2023; Lin et al., 2023). We randomly mask 15%, 30%, and 45% of joints as input to the masked pathway. As in previous findings (Zhu et al., 2023), marginal differences are observed between different masking ratios $< 50\%$ of the joints, with the highest accuracy being 89.7% at 15% RM (see Table 5). In comparison, we experiment with our TM approach by varying $\delta$ between 0.1-0.5 (step size = 0.1), with lower values meaning masking more activated joints (see Figure 5 (Left)). We observe that the highest accuracy of 91.5% is achieved at $\delta = 0.2$, which is **1.8** percentage points higher than the 15% RM.

**Effect of Hyperparameters.** We analyze the effect of $\mathcal{L}_{sc}$ weight $\alpha$ (Figure 5 (middle)) and $\mathcal{L}_{cc}$ weight $\beta$ (Figure 5 (right)) on the overall model performance. At $\delta = 0.2$, we experiment with adding $\mathcal{L}_{sc}$ and $\mathcal{L}_{cc}$ separately, achieving a top accuracy of 93.2% and 93.8% at $\alpha = 9$ and $\beta = 1$ respectively.

**Ablation on contrastive losses.** We experiment with separately and collectively applying $\mathcal{L}_{sc}$ and $\mathcal{L}_{cc}$ with RM and TM (Table 5.) While each loss individually contributes to a performance gain, using the two losses together results in **2.3** and **2.4** improvement in percentage points with RM and TM respectively. Figure 3 shows that combining the two losses results in significant improvement in robustness against noise.

Table 5: Ablation Experiments on NTU60-XSub.

| Component | $\mathcal{L}_{ce}$ | $\mathcal{L}_{sc}$ | $\mathcal{L}_{cc}$ | Accuracy |
|---|---|---|---|---|
| No Masking | ✓ | | | 88.7 |
| **15% RM** | ✓ | | | **89.7** |
| 30% RM | ✓ | | | 89.6 |
| 45% RM | ✓ | | | 89.4 |
| 15% RM | ✓ | ✓ | | 91.1 |
| 15% RM | ✓ | | ✓ | 91.9 |
| **15% RM** | ✓ | ✓ | ✓ | **92.0** |
| TM ($\delta = 0.1$) | ✓ | | | 89.9 |
| **TM ($\delta = 0.2$)** | ✓ | | | **91.5** |
| TM ($\delta = 0.3$) | ✓ | | | 91.4 |
| TM ($\delta = 0.2$) | ✓ | ✓ | | 93.2 |
| TM ($\delta = 0.2$) | ✓ | | ✓ | 93.8 |
| **TM ($\delta = 0.2$)** | ✓ | ✓ | ✓ | **93.9** |

## 5 CONCLUSION

In this paper, we introduce MaskCLR, a new training paradigm for robust skeleton-based action recognition. Concretely, MaskCLR encodes more information from input skeleton joints through the targeted masking of the most activated nodes. Further, a multi-level contrastive learning framework is introduced to contrast skeleton representations at the sample and class levels, forming a class-dissociated feature space that enhances the model accuracy, robustness to perturbations, and generalization to pose estimators. We demonstrate the effectiveness of our method on three popular benchmarks using different pose estimators, significantly outperforming existing works on standard and perturbed skeleton sequences.

**Limitation.** We find the most activated joints by using the attention weights from the final $\mathcal{ST}$ blocks only. Recent methods such as (Chefer et al., 2021) have shown that using the attention scores from multiple layers is better in finding the activated input regions. We leave this for future work.

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

# A APPENDIX

## A.1 DETAILS OF DATASETS

**NTU RGB+D.** NTU RGB+D (Shahroudy et al., 2016; Liu et al., 2019) is lab-collected, large-scale action recognition dataset which has two versions: NTU-60 (60 classes) and NTU-120 (120 classes.) NTU-60 contains 57K videos while NTU-120 is an extension of it that contains 114K videos. The datasets are split in three ways: X-Sub (for both), X-View (for NTU-60), and X-Set (for NTU-120), in which human subjects, camera views, and camera setups are different, respectively. While 3D skeletons from sensors are provided in this dataset, we extract the 2D skeletons by applying the three different pose estimators (see below) directly on the RGB videos.

**Kinetics400.** Kinetics400 (Kay et al., 2017) is a large scale video-based action recognition dataset with 400 action classes and 300k videos. The videos are 10s long extracted from YouTube which makes the dataset a challenging one due to the diversity in quality of videos, number of people, and background noise in each video. Table 6 summarizes the datasets utilized in our experiments.

Table 6: **Action recognition datasets.**

| Dataset | Source | #Classes | #Train | #Val. | Total |
|---|---|---|---|---|---|
| NTU60-XSub (Shahroudy et al., 2016) | Lab | 60 | 40K | 17K | 57K |
| NTU60-XView (Shahroudy et al., 2016) | Lab | 60 | 38K | 19K | 57K |
| NTU120-XSub (Liu et al., 2019) | Lab | 120 | 63K | 51K | 114K |
| NTU120-XSet (Liu et al., 2019) | Lab | 120 | 54K | 60K | 114K |
| Kinetics400 (Kay et al., 2017) | YouTube | 400 | 250K | 50K | 300K |

**Pose Extraction.** Pose estimation is a critical step that largely affects the final recognition accuracy, yet the importance of which is mostly overlooked in previous literature. Poses retrieved from sensor readings or existing pose estimators are used to train and test skeletal action recognition models without strong justification behind the pose extraction method. To the best of our knowledge, there's no consensus among the research community on a fixed set of skeletons to test action recognition performance. Furthermore, due to the large volume of research, it is not feasible to conduct a comprehensive study on which models work best on which poses and for which datasets. We, therefore, argue for the need for skeletal action recognition models that are generic to the type of pose estimator. We highlight the importance of reporting the model performance on poses extracted with multiple pose estimators instead of only one. To that end, we leverage three pose estimators of different levels of performance: ViTPose (SOTA) (Xu et al., 2022) (High Quality, HQ), HRNet (Sun et al., 2019) (Medium Quality, MQ), and OpenPifPaf (Kreiss et al., 2021) (Low Quality, LQ).

Table 7: **Quality of utilized pose estimators based on the AP score on COCO test-dev set.**

| Pose Estimator | Type | AP | Pose Quality |
|---|---|---|---|
| ViTPose (Xu et al., 2022) | Top-Down | 81.1 | HQ |
| HRNet (Sun et al., 2019) | Top-Down | 77.0 | MQ |
| OpenPifPaf (Kreiss et al., 2021) | Bottom-Up | 71.9 | LQ |

We leverage 2D poses instead of 3D ones because in general they are of higher quality (Duan et al., 2022b). As shown in Table 7, the selected pose estimators have different types and pose qualities, assigned according to their reported AP score on the COCO test-dev (Lin et al., 2014). While Top-Down methods outperform Bottom-Up methods on standard benchmarks, we highlight the importance of experimenting with both to demonstrate the generalization of skeletal action recognition. Following previous literature (Duan et al., 2022b), we store the extracted keypoints in the 17-joint coco format in coordinate triplets $(x, y, c)$, where $(x, y)$ is the joint coordinates and $c$ is the joint confidence score. In Table 8, we report some metrics reflecting the percentage of keypoints and people that were undetected by each pose estimator. The percentage of missing keypoints is the number of missed keypoints within the detected poses, divided by the actual number of joints in such poses. The percentage of missing people indicate the number of undetected people divided by the total number of people in each dataset.

Table 8: **Assessment of pose estimators in terms of undetected joints and people.**

| Pose Estimator | NTU60 | | NTU120 | | NTU60 | | NTU120 | |
| | XSub | XView | XSub | XSet | XSub | XView | XSub | XSet |
| --- | --- | --- | --- | --- | --- | --- | --- | --- |
| Metric | Percentage of missing keypoints | | | | Percentage of missing people | | | |
| ViTPose | 0.0 | 0.0 | 0.0 | 0.0 | 1.2 | 0.01 | 0.01 | 0.04 |
| HRNet | 0.0 | 0.0 | 0.0 | 0.0 | 0.1 | 0.1 | 0.12 | 0.13 |
| OpenPifPaf | 7.7 | 6.9 | 6.5 | 7.5 | 0.53 | 0.29 | 5.4 | 5.8 |

## A.2 ATTENTION FILTERS DIAGNOSTICS

We analyze the effect of finding the activated joints from different attention maps across three MHSA depth layers in the transformer network. Our goal is to find the most important joints that lead to the action classification. We follow the black-box insertion/deletion metric proposed in RISE (Petsiuk et al., 2018) for empirically evaluating the different attention maps. For the deletion metric, we incrementally delete the most important joints, as computed by the attention scores of a transformer layer, and measure the effect on the accuracy by computing the Area Under the Curve (AUC). On the other hand, the insertion metric is a complementary approach in which the most important joints are gradually introduced. Our results are shown in Figure 6 for different attention maps on NTU60-XSub (Shahroudy et al., 2016). The best attention map is determined by a lower deletion AUC and a higher insertion AUC. We find that the attention map from the last MHSA block $N = 5$ best reflects the most important joints. The last layer inherits information from all the preceding layers in learning attention parameters, and is therefore used in our approach to determine the most activated joints.

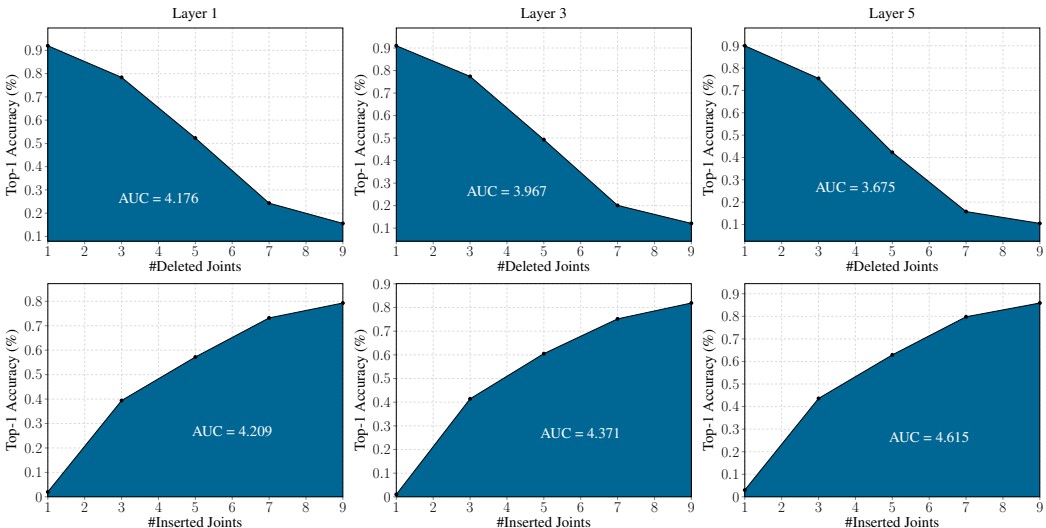

Figure 6: **Deletion (top) and insertion (bottom) metrics for attention maps of three layers.**

## A.3 NOISE VISUALIZATION

In Figure 7, we compare visualizations of standard and noisy skeletons. At noise $\sigma \leq 0.005$, we note that the resulting noisy skeletons are virtually indistinguishable from the original skeletons. Figure 8 shows the class-wise performance gained on noisy skeletons ($\sigma = 0.002$) from training with our MaskCLR approach. Particularly in low-motion actions (*e.g,* drink water, pointing, etc), MaskCLR obtains considerable performance gains, up to **28.4** percentage points for "pointing." MotionBERT only captures low-level joint motion patterns, which might be disrupted by noise. Conversely, MaskCLR encodes the pose information, which is important in differentiating fine-grained actions. Additionally, MaskCLR captures the high-level action semantics, which do not change under action-preserving levels of noise.

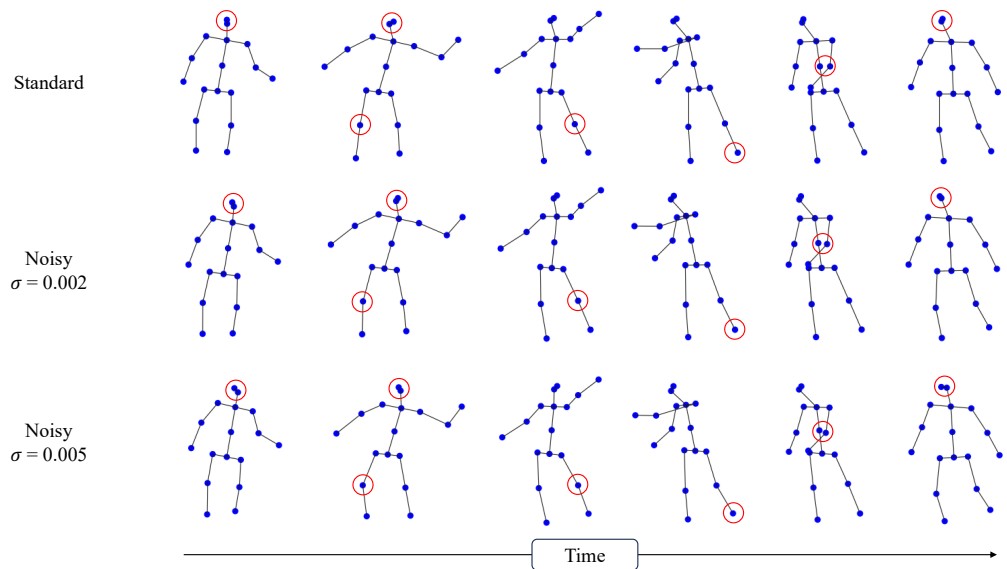

Figure 7: **Visualization of standard and noisy versions of action "throw" from NTU60-XSub.** Noise is sampled from a Gaussian Distribution $X \sim \mathcal{N}(0, \sigma^2)$ and introduced on all joints across time. At $\sigma = 0.002$, the noisy skeletons are virtually indistinguishable from the standard ones. The red circles reflect subtle differences in joint positions.

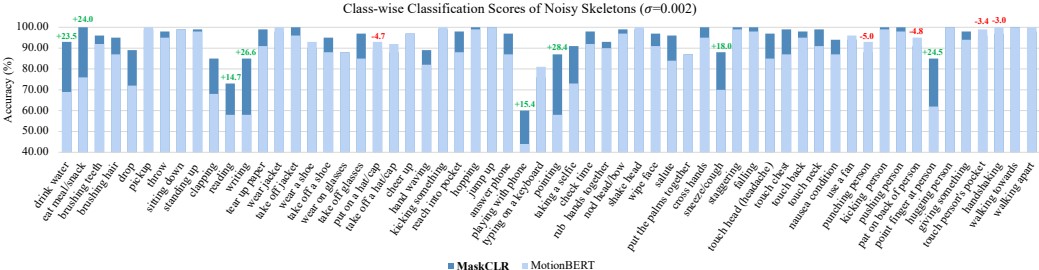

Figure 8: **Class-wise accuracy on NTU60-XSub under Gaussian noise** ($\sigma = 0.002$). MaskCLR improves the classification performance in most classes, especially in subtle actions such drink water, reading, writing, etc. Our approach exploits the pose information from previously unactivated joints to reduce the confusion between low-motion action classes (Best viewed in color.)

## A.4 MORE EXPERIMENTS

**Targeted masking.** In Figure 9, we compare the deletion AUC, proposed in RISE (Petsiuk et al., 2018), for baseline MotionBERT and MaskCLR based on the attention map of the last layer $N = 5$. We note that targeted masking is more challenging than random masking since the occluded joints are the ones that contribute most to the classification prediction. MaskCLR outperforms MotionBERT by **2.7** in deletion AUC. Our targeted masking strategy helps the model explore a bigger set of discriminative joints (as shown in Figure 1), thus alleviating the dependence on a few number of joints to recognize actions.

**Shifted Joints.** To further evaluate the robustness of our approach, we randomly shift different numbers of joints in the input skeleton sequence and report the effect on accuracy. More specifically, we shift 1, 3, 5, and 10 joints (selected randomly) in the input skeleton sequence to a random position within the skeleton bounding box. Shifted joints are commonly observed in the output of pose estimators (Kreiss et al., 2021; Xu et al., 2022; Sun et al., 2019). As shown in Figure 10, shifted joints cause rapid drop in the accuracy of SOTA methods. In contrast, MaskCLR exhibits the lowest

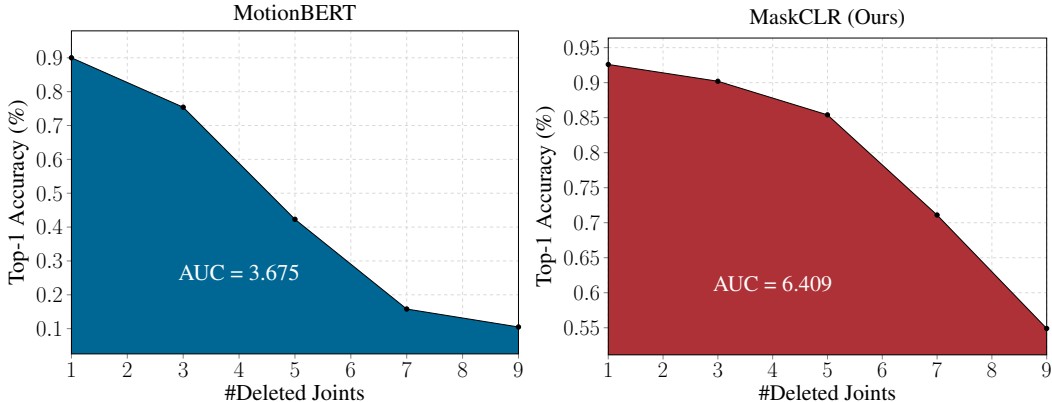

Figure 9: **Deletion score for baseline MotionBERT and MaskCLR.**

drop in accuracy, notably surpassing baselines MotionBERT by **18** percentage points when the number of shifted joints is 10.

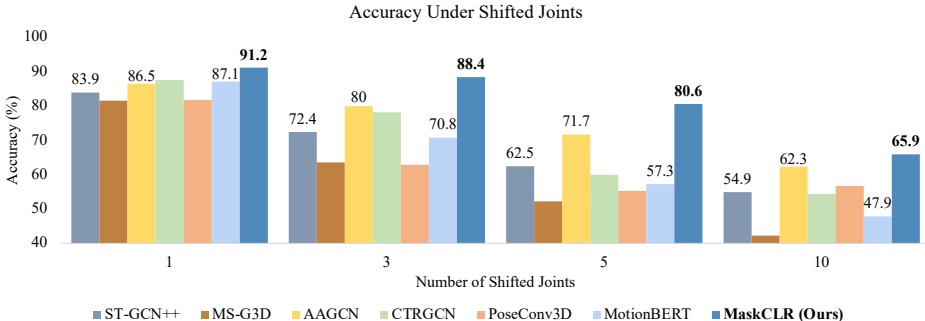

Figure 10: **Top-1 accuracy vs the number of shifted joints.**

## A.5 QUALITATIVE RESULTS

**Confusion Matrix.** In Figure 11, we visualize the confusion matrices of MotionBERT and MaskCLR on NTU60-XSub under spatiotemporal noise $\sigma = 0.005$. We observe that MotionBERT misclassifies most actions into high-motion classes such as "wear a jacket" and "one foot jumping." One possible explanation is that the introduced noise causes *artificial* movements in skeleton joints. While such fluctuations do not change the overall action semantics, it introduces motion to *all* joints, which typically happens in high-motion actions. Hence, the model misclassifies the sequence into a high-motion action. Focusing on low-level joint variations leads to the accuracy deterioration of MotionBERT under noisy skeletons. Instead, MaskCLR aims at capturing the high-level action semantics by utilizing a larger number of informative joints, the holistic motion of which do not change under small amounts of noise. Additionally, the rich cross-sequence intrinsic information shared between skeleton sequences of the same class is exploited through our multi-level contrastive learning approach. Consequently, MaskCLR is better able to handle perturbed skeleton sequences, as reflected in the confusion matrix (Figure 11.)

**t-SNE visualization of feature space.** Figure 12 shows the t-SNE visualizations of the feature space of NTU60-XSub before the final classification layer of MotionBERT and MaskCLR. We observe the feature space of our method is better disentangled across most classes, which we attribute to the added sample- and class-level contrastive losses.

Classification of noisy skeletons at $\sigma = 0.005$

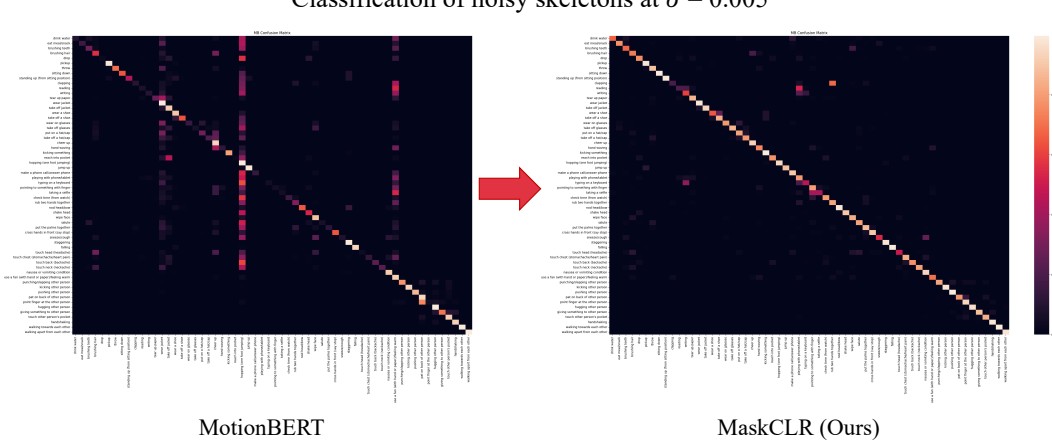

MotionBERT                          MaskCLR (Ours)

Figure 11: **Confusion matrices of noisy skeletons from NTU60-XSub.** MaskCLR reduces the ratio of false positives and false negatives by establishing clearer decision boundaries between representations of different classes in the feature space.

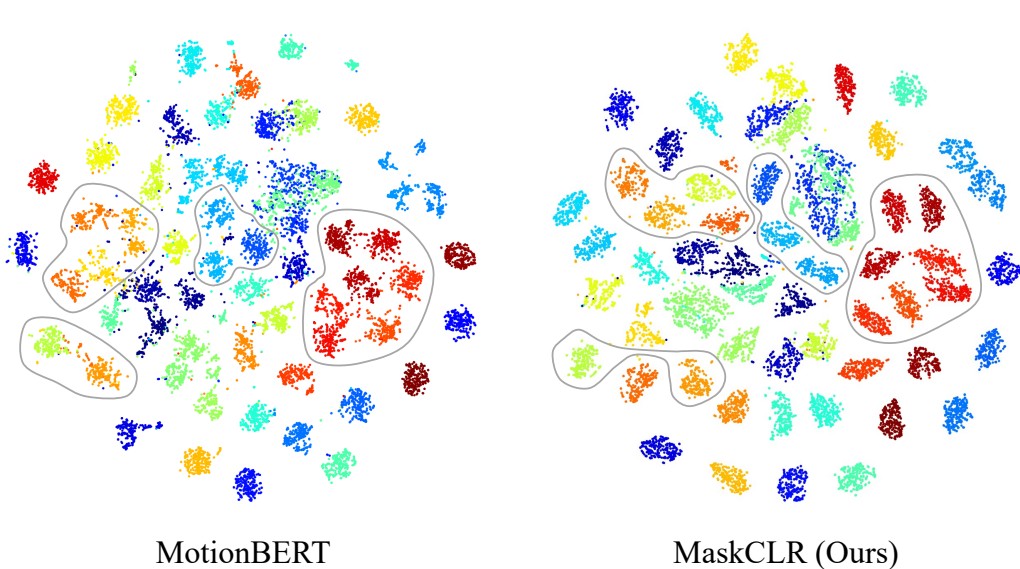

MotionBERT                          MaskCLR (Ours)

Figure 12: **t-SNE visualizations of feature space on NTU60-XSub.** The feature representations of our MaskCLR is better clustered and well-disentangled compared to that of MotionBERT. Our multi-level contrastive learning approach minimizes the distance between similar input skeletons at both the sample and class levels, boosting the robustness of the model against noisy or incomplete skeletons and improving the overall classification accuracy. (Best viewed in color.)

