# OpenReview forum: "MaskCLR: Multi-Level Contrastive Learning for Robust Skeletal Action Recognition"
_ICLR.cc/2024/Conference — ICLR 2024 Conference Withdrawn Submission_

### Official Review · Reviewer_WKXM · 2023-10-16

**Soundness:** 4 excellent
**Presentation:** 4 excellent
**Contribution:** 3 good
**Rating:** 5
**Confidence:** 4

**Summary:**

This paper provides a new supervised contrastive-learning method via key-point masking to explore richer features from inactivated key-points. The results demonstrate its better performance compared to SoTA skeleton-based action recognition approaches, especially in the robustness.

**Strengths:**

### Main strength points:
- Better performance, especially in the robustness, which makes the skeleton-based action recognition algorithms more deployable.
- Comprehensive ablations and experiments for the masking method.
- Good writing.
- Codes are available.

**Weaknesses:**

### Main weak point:
- The paper combine several **existing** methods to achieve better performance. Specifically, an additional path is added to improve the performance and extract more features. However, this duplicates the computational cost. How much performance gain can we get if we **simply use the upper path and make the ST-MSHA block larger**? Fairer comparison should be added under the scenario of **similar computational cost**.

I will be very willing to raise the score if the authors can address my concerns.

**Questions:**

Similar methods can also be applied to non-skeleton based methods. We know the skeleton-based methods are more robust than non-skeleton-based methods. But will that conclusion still hold when we add masking as mentioned in the paper?

---

> ### Author Response · Authors · 2023-11-20
> **Response to Reviewer WKXM**
>
> Thank you very much for these informative and constructive comments. We are delighted that you found our work valuable to the community in terms of making skeletal action recognition models more robust and more deployable.
>
> We would like to clarify some points regarding our method.
>
> > **P1:** How much performance gain can we get if we simply use the upper path and make the ST-MSHA block larger?
>
> **R:** Making ST-MSHA blocks larger could have multiple interpretations. This includes increasing the number of blocks (depth), increasing the input and output feature dimensions, increasing the number of heads, MLP ratio, etc. Such experimentation has already been conducted by the authors of MotionBERT [1] and is provided in their ablation and appendix sections. The reported accuracy is the one obtained after optimizing all of these parameters to obtain the highest accuracies on the different datasets. We recognize the importance of this point because optimizing these parameters does indeed significantly improve the model performance. Yet, we further improve the model performance using our training approach. Moreover, we boost the model robustness to perturbations such as occlusion and noisy and we improve generalization to pose estimators of different levels of quality.
>
>
> > **P2:** Fairer comparison should be added under the scenario of similar computational cost.
>
> **R:** Thank you for raising this point. As we mention in the paper, our method does not change the number of parameters of the backbone transformer model and does not change the number of FLOPs. Therefore, our comparisons are by default under the same computational cost at test time. However, at training time, our approach requires an extra amount of computation which varies from a model to the other. For MotionBERT, we train for twice the number of epochs of the original model, first training using cross entropy loss then with the combined contrastive approach discussed in the paper. We note that, generally speaking, the training time depends on the backbone model, and is less important than the inference cost, which does not change under our approach.
>
> > **P3:** I will be very willing to raise the score if the authors can address my concerns.
>
> **R:** Thank you very much for the kind gesture. We hope we addressed your concerns.
>
> > **P4:** Similar methods can also be applied to non-skeleton based methods. We know the skeleton-based methods are more robust than non-skeleton-based methods. But will that conclusion still hold when we add masking as mentioned in the paper?
>
> **R:** Indeed, this is a very interesting line of thought. Especially, for RGB-based method transformer-based methods work similarity, which could inspire future work in this domain. We hope that our work would be valuable for the community in other domains as well such as image classification, object detection, and trajectory prediction.
>
> [1] MotionBERT: A Unified Perspective on Learning Human Motion Representations. ICCV'23

---

### Official Review · Reviewer_vH9p · 2023-10-26

**Soundness:** 2 fair
**Presentation:** 3 good
**Contribution:** 2 fair
**Rating:** 3
**Confidence:** 5

**Summary:**

This paper tackles the task of skeleton action recognition. Through an analysis of the attention weights of MotionBERT, the authors highlight the model's concentration on a limited set of discriminative joints for action recognition. As a response to this observation, they introduce a novel approach, which involves masking out the most highly activated joint to encourage the model to explore a broader array of informative joints. The proposed method leverages two distinct contrastive loss functions at both sample and class levels. To assess its efficacy, the approach is evaluated on three widely recognized benchmark datasets.

**Strengths:**

1. The paper is well organized and easy to follow.
2. The paper presents a commendable analysis and visualization of the attention weights of MotionBERT, offering a deeper insight into the model's inner workings.
3. Comprehensive experiments are conducted on three benchmark datasets, also with the results on the robustness against skeleton perturbations.

**Weaknesses:**

1. In regard to motivation, the inspiration for this paper stemmed from an in-depth examination of the attention weights employed by MotionBERT. This analysis revealed that MotionBERT focuses on a restricted set of joints for recognition. However, this gave rise to two questions:

(a) The paper primarily centers its analysis and experiments on MotionBERT, leading to doubts about whether the identified issues are prevalent in various backbone architectures rooted in self-attention mechanisms.

(b) It is noteworthy that the proposed method can exclusively be applied to models built upon self-attention. This constraint, in turn, restricts the method's applicability and generalizability to non-transformer backbones.

2. In terms of novelty and contributions, the primary innovation presented in this paper involves the process of masking out the most highly activated joint and employing a contrastive loss function to reduce the dissimilarity between the original joint embedding and the masked counterpart. Although the idea is conceptually straightforward, in my view, its novelty might be somewhat limited for publication at ICLR.

3. About literature review: it's worth noting that certain recent works on skeleton action recognition have been omitted, such as [1-5].

4. About experimental evaluation: it's notable that the authors only present the results for the joint modality. However, it is a standard practice in the field to present results for the multi-stream fusion approach, which encompasses joint, bone, joint motion, and bone motion modalities, as demonstrated in previous studies [1-5]. Furthermore, it's essential to acknowledge that the performance of the proposed method falls short when compared to PoseConv3D on NTU120-XSet and Kinetics400 datasets. On NTU60-XView, the improvement is only marginal.

[1] Huang, et al. "Graph contrastive learning for skeleton-based action recognition." ICLR 2023.

[2] Lee, et al. "Hierarchically decomposed graph convolutional networks for skeleton-based action recognition." ICCV 2023.

[3] Lee, et al. "Leveraging spatio-temporal dependency for skeleton-based action recognition." ICCV 2023.

[4] Wang, et al. "Neural Koopman Pooling: Control-Inspired Temporal Dynamics Encoding for Skeleton-Based Action Recognition." CVPR 2023.

[5] Foo, et al. "Unified pose sequence modeling." CVPR 2023.

**Questions:**

None

---

> ### Author Response · Authors · 2023-11-20
> **Response to Reviewer vH9p (Part 1/2)**
>
> We appreciate your feedback and comments which are paramount in the process of refining and polishing this work. We find your reviews very insightful and speak to the core of this work. Thank you very much for the time and effort you put into reading this paper carefully.
>
> Below we discuss your concerns on our proposed framework.
>
> > **P1:** (a) The paper primarily centers its analysis and experiments on MotionBERT, leading to doubts about whether the identified issues are prevalent in various backbone architectures rooted in self-attention mechanisms.
>
> **R:** Thank you for this important observation. The reason why we chose MotionBERT is that it’s the best-performing transformer-based model whose implementation is ‘publicly available’ at the time of the start of this work. MotionBERT core design is very similar to other transformer-based methods in the sense that it uses spatial and temporal MHSA blocks. How these blocks are ordered and fused together is different from method to another, but virtually in all of them you can conveniently compute the attention scores given to every keypoint and hence, you can apply the idea of targeted masking and MLCL quite easily. In support of this statement, and to address your concern about the applicability of our method to other transformer-based backbones, we conduct further experiments on two new transformers: the vanilla transformer [1] and STTFormer [2]. On the vanilla transformer, we apply SkeletonMAE framework (originally proposed for GCN-based methods) using the vanilla transformer backbone. We provide the results below:
>
> **Table 1: Performance improvement (%) compared to transformer-based backbones. Parentheses indicate improvement over the second best method.**
>
> | Method          | Backbone      | NTU60-XSub | NTU60-XView | NTU120-XSub | NTU120-XSet |
> |-----------------|---------------|------------|-------------|-------------|-------------|
> | AimCLR [3]      | STTFormer     | 83.9       | 90.4        | 74.6        | 77.2        |
> | CrossCLR [4]    | STTFormer     | 84.6       | 90.5        | 75.0        | 77.9        |
> | SkeletonMAE [5] | STTFormer     | 86.6       | 92.9        | 76.8        | 79.1        |
> | MaskCLR (Ours)  | STTFormer     | 90.1 (**+ 3.5**)       | 95.4 (**+ 2.5**)       | 79.0      (**+ 2.2**)  | 80.5   (**+ 1.4**)     |
> | SkeletonMAE [5] | Transformer   | 88.5       | 94.7        | 87.0        | 88.9        |
> | MAMP [6]        | Transformer   | 93.1       | 97.5        | 90.0        | 91.3        |
> | MaskCLR (Ours)  | Transformer   | 93.5    (**+ 0.4**)   | 97.5        | 90.5   (**+ 0.5**)     | 91.9    (**+ 0.6**)    |
>
> As shown, MaskCLR consistently improves the performance of the other transformer-based methods, showing the generalization and efficacy of our approach. We conclude the repones to this point by emphasizing that our focus is not on MotionBERT as an architecture, but on the idea of self-attention which is used in virtually all transformer-based methods.
>
> > **P2:** (b) It is noteworthy that the proposed method can exclusively be applied to models built upon self-attention. This constraint, in turn, restricts the method's applicability and generalizability to non-transformer backbones.
>
> **R:** That’s exactly correct. We present this approach for transformer-based methods as discussed in the paper. The reason is that we take advantage of the attention weights computed by the self-attention mechanism to empirically find the most important joints for targeted masking. However, it’s important to note that the same idea could be applied to other backbones with a slightly different approach. For example, for convolution-based methods such as GCNs and 3D-CNNs, different other techniques could be employed to find the most important joints. CAM [7] and Grad-CAM [8] are some quick and easy-to-apply methods that are proposed to find the saliency regions in images but could be slightly tweaked to be used in the skeleton domain. In doing so, the rest of our approach, i.e., TM and MLCL, could be conveniently applied for convolution-based methods. We leave this for future work.
>
> [1] An image is worth 16x16 words: Trans- formers for image recognition at scale. arXiv'20
>
> [2] Spatiotemporal tuples transformer for skeleton-based action recognition. arXiv'22
>
> [3] Contrastive learning from extremely augmented skeleton sequences for self-supervised action recognition. AAAI'22
>
> [4] CrossCLR: Cross-modal contrastive learning for multi-modal video representations. ICCV'21
>
> [5] SkeletonMAE: Graph-based Masked Autoencoder for Skeleton Sequence Pre-training, ICCV'23
>
> [6] Masked motion predictors are strong 3d action representation learners. ICCV'23
>
> [7] Learning Deep Features for Discriminative Localization. CVPR'16
>
> [8] Grad-CAM: Visual Explanations from Deep Networks via Gradient-based Localization. ICCV'17

---

> > ### Author Response · Authors · 2023-11-20
> > **Response to Reviewer vH9p (Part 2/2)**
> >
> > > **P3:** In terms of novelty and contributions, the primary innovation presented in this paper involves the process of masking out the most highly activated joint and employing a contrastive loss function to reduce the dissimilarity between the original joint embedding and the masked counterpart. Although the idea is conceptually straightforward, in my view, its novelty might be somewhat limited for publication at ICLR.
> >
> > **R:** We respectfully disagree with you in this point. To the best of our knowledge, MaskCLR is the first work to tackle the issue of robustness in transformer-based methods, and the first to address the issue of generalization to pose estimators in all skeletal action recognition methods in general. Our work proposes a new masking technique that is guided by the attention weights from transformer models, which has empirically been shown to result in performance improvements on all benchmarks. This is the first work to highlight this observation and effectively take advantage of it with the MLCL approach to learn the high-level actions semantics instead of low-level joint variations. Finally, we validate our approach with plenty of experiments and ablation studies on three different benchmarks, with three different pose estimators, and under different forms of skeleton perturbations. Our method shows consistent improvements under almost all of the aforementioned cases, which demonstrates and efficacy and generalization of our idea.
> >
> >
> > > **P4:** About literature review: it's worth noting that certain recent works on skeleton action recognition have been omitted, such as [1-5].
> >
> > **R:** Thank you for pointing this out. The reference you provided are indeed very informative and relevant to our work. We will update the related works section accordingly.
> >
> > > **P5:** About experimental evaluation: it's notable that the authors only present the results for the joint modality. However, it is a standard practice in the field to present results for the multi-stream fusion approach, which encompasses joint, bone, joint motion, and bone motion modalities, as demonstrated in previous studies [1-5]. Furthermore, it's essential to acknowledge that the performance of the proposed method falls short when compared to PoseConv3D on NTU120-XSet and Kinetics400 datasets. On NTU60-XView, the improvement is only marginal.
> >
> > **R:** We are familiar with the multi-stream fusion approach. However, in the specific contribution of our work, the multi-stream fusion approach does not add much new information that contributes to our message. We believe it’s better to give more focus to demonstrating the robustness to different perturbations or generalization to pose estimators (or both), which are the main major improvements we highlight in this paper. We appreciate your point on multi-stream fusion as an important addition to compare with previous work. However, since it takes extra compute and time resources, we leave this for future work.
> >
> > As for performance comparisons, we do indeed put the PoseConv3D numbers in bold in the table 2, showing that its performance is higher on NTU120-XSet and Kinetics400. However, we present this work as a method to improve the performance of transformer-based backbones. Therefore, we are less concerned with surpassing SOTA records and more interested in improving the performance relative to the backbone transformer model. Notably, our method outperforms PoseConv3D using a different transformer backbone [1] as shown in the above results on NTU120-XSub and NTU120-Xset (as shown in the table above). Finally, given that the NTU dataset is an extremely popular and well-benchmarked dataset, we want to point out that even an absolute improvement of 0.2% is significant on NTU.  The average improvement we achieve across the three transformer backbones is to 1.6, 0.9, 1.8, and 1.7 percentage points on NTU60-XSub, NTU60-XView, NTU120-XSub, and NTU120-XSet respectively. We believe these improvements are quite significant in such a competitive field.
> >
> > [1] An image is worth 16x16 words: Trans- formers for image recognition at scale. arXiv'20

---

### Official Review · Reviewer_8Cc6 · 2023-10-26

**Soundness:** 3 good
**Presentation:** 3 good
**Contribution:** 3 good
**Rating:** 6
**Confidence:** 2

**Summary:**

The manuscript observes that current transformer-based skeletal action recognition methods tend to focus on a limited set of joints and low-level motion patterns, potentially resulting in performance degradation because of the action perturbation or ambiguousness shown in Fig.1. Therefore, the authors propose a Targeted Masking strategy to occlude the joints with highest activations, and a Multi-Level Contrastive Learning framework to push the original and altered feature embeddings (positive pair) together. Extensive experiments are conducted on NTU60, NTU120, and Kinetics datasets to verify the effectiveness, generalization, and robustness.

**Strengths:**

a. The motivation is well justified.
b. Experiments are sufficient to demonstrate the effectiveness and robustness.
c. The proposed MaskCLR reaches a new state-of-the-art on the benchmarks in use.

**Weaknesses:**

a.There is potential room for improvement in the writing skills. For example, in the first sentence of the Abstract, it may be more suitable to use "tend to focus on" instead of "focus on" when discussing existing methods. Given that the critique of previous methods largely stems from qualitative investigations presented in Fig. 1, it would be advantageous to acknowledge the potential influence of individual bias on these findings. Addressing this issue has the potential to enhance the overall quality of the paper.
b. Activation-guided augmentation is a concept relatively familiar within the vision community. However, it would be valuable to provide a more direct and explicit explanation of this point. Moreover, it may be better to probe some related works in this field in Sec.2, such as [1-2].

[1] He J, Li P, Geng Y, et al. FastInst: A Simple Query-Based Model for Real-Time Instance Segmentation[C]//Proceedings of the IEEE/CVF Conference on Computer Vision and Pattern Recognition. 2023: 23663-23672.
[2] Choe J, Shim H. Attention-based dropout layer for weakly supervised object localization[C]//Proceedings of the IEEE/CVF Conference on Computer Vision and Pattern Recognition. 2019: 2219-2228.

**Questions:**

Consider replacing the term 'multi-level' in the title with 'activation-guided' or similar terminology to better align with the core theme of this manuscript.
Additionally, it's worth noting that most related works in the field of contrastive action recognition are self-supervised and primarily evaluated using unsupervised metrics. However, it's important to point out that despite the reviewer's familiarity with supervised contrastive learning, there was still some confusion experienced until the end of Section 2. This suggests that there may be a need for the authors to dedicate more time to refining and clarifying their paper.
If the reviewer's understanding is generally accurate, it underscores the importance of improving the paper's clarity and readability.

---

> ### Author Response · Authors · 2023-11-20
> **Response to Reviewer Reviewer 8Cc6**
>
> We appreciate the time you invested and reading our paper and providing valuable insights that further contribute to the strength of our work. We are glad that you found our work good in terms of soundness, presentation, and contribution.
>
> Below we address some of your concerns, which we take into consideration to update our paper.
>
> > **P1:** There is potential room for improvement in the writing skills. For example, in the first sentence of the Abstract, it may be more suitable to use "tend to focus on" instead of "focus on" when discussing existing methods.
>
> **R:** Thank you very much. Indeed "tend to focus on" is a better expression in this case. We will update the paper accordingly.
>
> > **P2:** Given that the critique of previous methods largely stems from qualitative investigations presented in Fig. 1, it would be advantageous to acknowledge the potential influence of individual bias on these findings. Addressing this issue has the potential to enhance the overall quality of the paper.
>
> **R:** We appreciate you raising this point. Indeed, it’s an important aspect that need to be discussed and clarified. In the introduction section, we provide some visualizations and observations to motivate our idea. However, ‘individual bias’ is not the basis on which we develop our work. In the appendix section, particularly in sections A2 and A4, we offer quantitative examination that further validates our observations by reporting the insertion and deletion scores, proposed in RISE [1]. More specifically, we conduct an in-depth experimentation to study the effect of having big dependency on a few set of joints on the performance of the model on the overall NTU60-XSub dataset. For example, in Figure 9 in the appendix, we can see that starting from standard accuracy of 92.8%, removing just 3/17 of the most important joints results in dropping the accuracy to 75.4% and removing 5/17 damages the performance further to 42.3%, which reflects the huge dependency on a few number of joints. In comparison, we see more stable performance in MaskCLR (figure 9 in the appendix), where removing 3/17 and 5/17 joints results in 90.2% and 85.4% in accuracy consecutively. Our motivation is therefore, not just based on qualitative inspection but a quantitative examination of the model using different attention layers and performance evaluations on the entire dataset.
>
>
> > **P3:** b. Activation-guided augmentation is a concept relatively familiar within the vision community. However, it would be valuable to provide a more direct and explicit explanation of this point. Moreover, it may be better to probe some related works in this field in Sec.2, such as [1-2].
>
> **R:** Thank you for this valuable addition. We will take this into consideration in updating our manuscript.
>
> > **P4:** Consider replacing the term 'multi-level' in the title with 'activation-guided' or similar terminology to better align with the core theme of this manuscript.
>
> **R:** This an interesting suggestion. The reason why we termed our technique ‘muli-level’ is because we use contrastive learning at different levels (sample- and class- levels) as described in the paper.  We think that 'activation-guided' would be an accurate term to describe our targeted masking strategy since it is indeed ‘activation-guided.’
>
>
> > **P5:** it's important to point out that despite the reviewer's familiarity with supervised contrastive learning, there was still some confusion experienced until the end of Section 2.
>
> **R:** Your feedback is quite valuable to us. We will revise the paper with more focus on section 2 and before.
>
> [1] Rise: Randomized input sampling for explanation of black-box models. arXiv, 2018.

---

> > ### Comment · Reviewer_8Cc6 · 2023-11-21
> >
> > Thank you for your thoughtful response, which has addressed some of my concerns.
> > From the reviewer's standpoint, the multi-level design appears to be an incremental contribution, whereas the empirical investigation depicted in Fig. 1, along with the resulting activation-guided contrastive learning, presents a more novel aspect in the realm of skeletal action recognition.
> > For this reason, the reviewer would like to suggest discussing more related works about activation-guided augmentation (not limited to the field of action recognition)  and reorganizing the title.
> > It is acknowledged that the author may hold a different perspective, and the reviewer is open to considering alternative viewpoints.
> > The reviewer is inclined to assign a rating marginally above the borderline for this paper.
> > Given the contribution of introducing novel aspects to action recognition, the reviewer maintains the original rating.

---

### Official Review · Reviewer_VBCi · 2023-10-29

**Soundness:** 2 fair
**Presentation:** 2 fair
**Contribution:** 2 fair
**Rating:** 3
**Confidence:** 3

**Summary:**

This paper introduces a masked contrastive learning approach for skeleton-based action recognition.

The approach employs a targeted masking strategy to obscure significant joints, thereby encouraging the model to learn a broader set of discriminative joints.

Additionally, a multi-level contrastive learning framework is proposed to enhance the feature embeddings of skeletons. This serves the dual purpose of ensuring compactness within each class and increasing dispersion among different classes.

Extensive experiments are conducted on various benchmarks to demonstrate the improvements over existing methods.

**Strengths:**

+ Overall the paper is technical sound.

+ Some nice visualizations presented in the paper.

+ Extensive experimental results and comparisons, ablation studies are well presented.

**Weaknesses:**

Major:

- The novelty of this work is limited.

(i) Transformer-based models are not thoroughly discussed and analyzed (which are closely related to the argument 'seamlessly incorporated into transformer-based models' as mentioned in abstract section. For example, [A].

[A] Focal and Global Spatial-Temporal Transformer for Skeleton-based Action Recognition, ACCV'22

(ii) Why only choose MotionBERT? The reviewer noticed that recent works, e.g., [B] and [C], are not discussed in the paper. There are only 3 works referenced from 2023, why?

[B] 3Mformer: Multi-Order Multi-Mode Transformer for Skeletal Action Recognition, CVPR'23

[C] SkeletonMAE: Graph-based Masked Autoencoder for Skeleton Sequence Pre-training, ICCV'23

(iii) The reviewer noticed that [B] also uses a very similar two pathways setup, the pipeline looks quite similar? (why it is not discussed and it seems that they also use the same dataset for evaluation, and the differences are (1) they use skeletal hypergraph whereas this work uses skeletal graph (2) they use standard classification loss whereas this work applies contrastive learning, so the novelty is the introduced targeted masking strategy?) Also it is a transformer-based model that uses hypergraph? The reviewer noticed that [B] also considers unactivated but informative joints?

(iv) Furthermore, [B] is based on a transformer architecture that utilizes hypergraphs. The question arises as to whether the proposed strategy can be seamlessly integrated into their transformer-based model.

- The justifications of design choice/principle and rationale are not properly illustrated. Why the model are arranged/designed in this way, and its links/relationships to related works are not provided and discussed.

- The maths presented in the paper is confusing and some terms are not explained clearly and properly.

(i) It is suggested to have a notation section for the maths symbols. Why the $_b$ in Eq (1) and (2) sometimes in bold face, it is quite confusing.

(ii) What is the relationship among $\textbf{A}$, $\textbf{A}\_b$, $\textbf{A}\_{\mathcal{S}}$ and $\textbf{A}\_{\mathcal{T}}$. Why $\textbf{A} = 1/2 * (\textbf{A}\_{\mathcal{S}} + \textbf{A}\_{\mathcal{T}})$, and '*' denotes multiplication? The design choice/principle and rationale are not properly illustrated.

(iii) In Eq. (4) $\bigotimes$ is not explained. The reviewer noticed that in Fig. 2, it is mentioned but it would be much clearer to have that in main texts.

- Table 2, the comparisons w.r.t. transformer-based (only 1 method is compared) are very limited. Table 4, the red highlighted results are not discussed and analyzed.

- The authors show Fig. 3 and 4, but the discussions and analysis w.r.t. different perspectives / new insights are very limited. It is suggested to add more discussions and analysis.

- The experiments w.r.t. whether the proposed strategy can be applied to transformer-based models are very limited (only 1 model is experimented). It is suggested to explore more transformer-based models.


Minor:

- Figure 1 is not very clear to reviewer. What dataset does these videos come from? The color is also not very visible and clearer enough to reviewer. The explanations are very limited.

- Figure 2 some fonts are too smaller to read.

**Questions:**

Please refer to weakness section.

More questions:

- Page 2, 'multi-level contrastive learning' part, what does this 'class-averaged features' mean?

- In contribution section of Introduction, how 'unactivated' and 'informative' joints are determined and to what extent?

---

> ### Author Response · Authors · 2023-11-20
> **Response to Reviewer VBCi (Part 1/4)**
>
> Thank you for your precious time that you put in reading our paper carefully. We are delighted that you understood the core of our work, and you found our contributions technically sound and the experiments and ablations extensive and demonstrative of performance improvements.
>
> We find your feedback deep and constructive, and below we address your concerns on our work:
>
> >**P1:** The novelty of this work is limited.
>
> **R:** We respectfully disagree with you in this point. To the best of our knowledge, MaskCLR is the first work to tackle the issue of robustness in transformer-based methods, and the first to address the issue of generalization to pose estimators in all skeletal action recognition methods in general. Our work proposes a new masking technique that is guided by the attention weights from transformer models, which has empirically been shown to result in performance improvements on all benchmarks. This is the first work to highlight this observation and effectively take advantage of it with the MLCL approach to learn the high-level actions semantics instead of low-level joint variations. Finally, we validate our approach with plenty of experiments and ablation studies on three different benchmarks, with three different pose estimators, and under different forms of skeleton perturbations. Our method shows consistent improvements under almost all of the aforementioned cases, which demonstrates and efficacy and generalization of our idea.
>
> >**P2:** Transformer-based models are not thoroughly discussed and analyzed (which are closely related to the argument 'seamlessly incorporated into transformer-based models' as mentioned in abstract section. For example, [A].
>
> **R:** We appreciate providing the reference for FG-STFormer. We believe it’s a relevant and recent transformer-based work that will further contribute to the comprehensiveness of our related works section. We do provide a discussion of the core idea of transformer-based methods in the methods section of our paper, which mainly targets Multi-Head Self-Attention (MHSA) that is used in virtually all transformer methods. The reason why we did not want to delve deeper in the specific design choices of different transformers is that this is not the main focus of our contribution. While our method mainly targets transformer-based methods, it is agnostic to the specific design choices of such method.

---

> ### Author Response · Authors · 2023-11-20
> **Response to Reviewer VBCi (Part 2/4)**
>
> >**P3:** (ii) Why only choose MotionBERT? The reviewer noticed that recent works, e.g., [B] and [C], are not discussed in the paper. There are only 3 works referenced from 2023, why?
>
> **R:** Thank you for this important observation. The reason why we chose MotionBERT is that it’s the best-performing transformer-based model whose implementation is ‘publicly available’ at the time. MotionBERT core design is very similar to other transformer-based methods in the sense that it uses spatial and temporal MHSA blocks. How these blocks are ordered and fused together is different from method to another, but virtually in all of them you can conveniently compute the attention scores given to every keypoint and hence, you can apply the idea of targeted masking and MLCL quite easily. In support of this statement, and to address your concern about the applicability of our method to other transformer-based backbones, we conduct further experiments on two other transformer backbones: the vanilla transformer [1] and STTFormer [2]. On the vanilla transformer, we apply SkeletonMAE framework (originally proposed for GCN-based methods). We could not, however, experiment with 3Mformer because it does not have an open-source implementation. We provide the results below:
>
> **Table 1: Performance improvement (%) compared to transformer-based backbones. Parentheses indicate improvement over the second best method sharing the same backbone.**
>
> | Method          | Backbone      | NTU60-XSub | NTU60-XView | NTU120-XSub | NTU120-XSet |
> |-----------------|---------------|------------|-------------|-------------|-------------|
> | AimCLR [3]      | STTFormer     | 83.9       | 90.4        | 74.6        | 77.2        |
> | CrossCLR [4]    | STTFormer     | 84.6       | 90.5        | 75.0        | 77.9        |
> | SkeletonMAE [5] | STTFormer     | 86.6       | 92.9        | 76.8        | 79.1        |
> | MaskCLR (Ours)  | STTFormer     | 90.1 (**+ 3.5**)       | 95.4 (**+ 2.5**)       | 79.0      (**+ 2.2**)  | 80.5   (**+ 1.4**)     |
> | SkeletonMAE [5] | Transformer   | 88.5       | 94.7        | 87.0        | 88.9        |
> | MAMP [6]        | Transformer   | 93.1       | 97.5        | 90.0        | 91.3        |
> | MaskCLR (Ours)  | Transformer   | 93.5    (**+ 0.4**)   | 97.5        | 90.5   (**+ 0.5**)     | 91.9    (**+ 0.6**)    |
>
> As shown, MaskCLR consistently improves the performance of the other transformer-based methods, showing the generalization and efficacy of our approach. We conclude the repones to this point by emphasizing that our focus is not on MotionBERT as an architecture, but on the idea of self-attention which is used in virtually all transformer-based methods.
>
>
> [1] An image is worth 16x16 words: Trans- formers for image recognition at scale. arXiv'20
>
> [2] Spatiotemporal tuples transformer for skeleton-based action recognition. arXiv'22
>
> [3] Contrastive learning from extremely augmented skeleton sequences for self-supervised action recognition. AAAI'22
>
> [4] CrossCLR: Cross-modal contrastive learning for multi-modal video representations. ICCV'21
>
> [5] SkeletonMAE: Graph-based Masked Autoencoder for Skeleton Sequence Pre-training, ICCV'23
>
> [6] Masked motion predictors are strong 3d action representation learners. ICCV'23

---

> ### Author Response · Authors · 2023-11-20
> **Response to Reviewer VBCi (Part 3/4)**
>
> >**P4:** (iii) The reviewer noticed that [B] also uses a very similar two pathways setup, the pipeline looks quite similar? (why it is not discussed and it seems that they also use the same dataset for evaluation, and the differences are (1) they use skeletal hypergraph whereas this work uses skeletal graph (2) they use standard classification loss whereas this work applies contrastive learning, so the novelty is the introduced targeted masking strategy?) Also it is a transformer-based model that uses hypergraph? The reviewer noticed that [B] also considers unactivated but informative joints?
>
> **R:**  The 3Mformer paper is a very interesting contribution and offers an excellent opportunity to be combined with our approach. We would like to highlight the differences between the contributions of 3Mformer and MaskCLR:
> 1. We would like to note that the contribution of 3Mformer is basically a new transformer architecture, unlike our proposed method which is a new framework for training existing transformer-based methods. Consequently, while they use a two-pathway approach in one part of their model, it is quite different from the one we propose in our work. In our framework, the two pathways share the same weights (basically the same model) with different inputs that effectively enrich the learning of information from the input skeletons. In 3Mformer, the two pathways have different weights but the same input (which is the opposite of what we propose.)
> 2. We propose to use the attention weights to guide the masking strategy. 3Mformer is employing a significantly different approach with no masking.
> 3. We propose a contrastive learning approach, as you indicated, which is quite different from 3Mformer.
> 4. It is not clear to us how 3Mformer considers the unactivated joints. While it is reasonable to assume that the usage of hypergraphs would cover more joints and higher order correlations, our work is different in the explicit targeting of the unactivated joints. To the best of our knowledge, our work is the first to take advantage of this technique for skeletal action recognition.
>
> >**P5:** (iv) Furthermore, [B] is based on a transformer architecture that utilizes hypergraphs. The question arises as to whether the proposed strategy can be seamlessly integrated into their transformer-based model.
>
> **R:** The 3Mformer is based on HoT (Higher Order Transformers) which essentially adopts higher-order self-attention to compute attentions weights of joints and hyper-edges at different orders 1,2, …, r. Since the higher-order attention maps reflect the importance of each joint (or group of joints in a hyperedge), our framework can be incorporated on top of 3Mformer in specific and hypergraph-based transformers in general by the targeted masking of the most important joints. Although the implementation of 3Mformer is not available, we will make sure to include hypergraph-based methods as part of our future research and experiments.
>
> >**P6:** The justifications of design choice/principle and rationale are not properly illustrated. Why the model are arranged/designed in this way, and its links/relationships to related works are not provided and discussed.
>
> **R:** We would like to reiterate that the design of the model itself is not the contribution of our work. Our contribution is the proposed framework of training the existing transformer-based models. For further details about the design choices of the model, please refer to the original paper of MotionBERT: https://arxiv.org/pdf/2210.06551.pdf
>
> >**P7:** (i) It is suggested to have a notation section for the maths symbols. Why the b in Eq (1) and (2) sometimes in bold face, it is quite confusing.
>
> **R:** Thank you for highlighting this. We will take your feedback into consideration and fix some of the notations for easier readability.
>
> **P8:** (ii) What is the relationship among, \( \mathbf{A} \), \( \mathbf{A}_{b} \), \( \mathbf{A}_{\mathcal{S}} \), and \( \mathbf{A}_{\mathcal{T}} \)? Why \( \mathbf{A} = \frac{1}{2} \cdot (\mathbf{A}_{\mathcal{S}} + \mathbf{A}_{\mathcal{T}}) \), and '*' denotes multiplication? The design choice/principle and rationale are not properly illustrated.
>
> **R:** Here, the idea is to average the spatial and temporal attention weights from the spatial and temporal MHSA blocks. \( \mathbf{A}_{\mathcal{S}} \) is the spatial attention scores, \( \mathbf{A}_{\mathcal{T}} \) is temporal attention scores, \( \mathbf{A}_{b} \) refers to both spatial and temporal scores, \( \mathbf{A} \) is the average of spatial and temporal scores. Thank you for pointing this out. While such definitions are all in the paper next to corresponding notations, we will consider having a notations section to further avoid any confusion.

---

> > ### Author Response · Authors · 2023-11-20
> > **Response to Reviewer VBCi (Part 4/4)**
> >
> > > **P9:** (iii) In Eq. (4)  is not explained. The reviewer noticed that in Fig. 2, it is mentioned but it would be much clearer to have that in main texts.
> >
> > **R:** Thank you for this suggestion. We will put it in the main text.
> >
> > > **P10:** Table 2, the comparisons w.r.t. transformer-based (only 1 method is compared) are very limited. Table 4, the red highlighted results are not discussed and analyzed.
> >
> > **R:** We believe this point has been addressed in our response above.
> >
> > > **P11:** The authors show Fig. 3 and 4, but the discussions and analysis w.r.t. different perspectives / new insights are very limited. It is suggested to add more discussions and analysis.
> >
> > **R:** It would be helpful if you can elaborate on the specific new insights that need to be discussed. In figures 3 & 4, the message is that our proposed framework boosts the model robustness against perturbations such as spatial and spatiotemporal noise (figure 3) and joint and part masking (figure 4.) Our targeted masking strategy expands the set of discriminative joints and our MLCL idea contributes to the formation of a clustered feature space, where different classes are pushed away, clearing better decision boundaries for classification. Hence, MaskCLR encourages the model to capture the high-level action semantics instead of low-level variations and is, therefore, less vulnerable to perturbations such the ones we present in Figures 3 & 4. We provide further insights and analysis in the Appendix (such as feature space visualizations and class-wise classification performance.) Kindly let us know what new insights would better strengthen this aspect of our work. We really appreciate your input and suggestions.
> >
> > > **P12:** The experiments w.r.t. whether the proposed strategy can be applied to transformer-based models are very limited (only 1 model is experimented). It is suggested to explore more transformer-based models.
> >
> > **R:** We believe this point has been addressed in our response above.
> >
> > > **P13:** Figure 1 is not very clear to reviewer. What dataset does these videos come from? The color is also not very visible and clearer enough to reviewer. The explanations are very limited.
> >
> > **R:** The dataset is NTU60-XSub. Thank you for pointing this out. We will clarify this in the main text. The qualitative analysis from Figure 1 shows that MotionBERT uses a small set of discriminative joints to recognize different actions, while the other unactivated joints do carry useful information that could aid in action classification. That’s the main motive behind our idea of expanding the set of discriminative joints by masking these activated joints and encouraging the model to recognize the actions in the absence of these joints. As shown in Figure 1, MaskCLR does indeed focus more on the other informative joints and the results, therefore, reflect improved performance in comparison to baseline MotionBERT.
> >
> > > **P14:** Figure 2 some fonts are too smaller to read.
> >
> > **R:** Thank you, we will make the font bigger for figure 2.
> >
> > > **P15:** Page 2, 'multi-level contrastive learning' part, what does this 'class-averaged features' mean?
> >
> > **R:** The class-averaged features are the average representations of features that correspond to a specific class. For example, for class l, the class averaged features are:
> >
> > C^l = (Σ R^l) / G^l
> >
> > Where R^l is a feature representation of a sample with ground truth l, and G^l is the number of representations that share the ground truth label l. Please refer to Eq. 6 that explain the class-averaged features.
> >
> > > **P16:** In contribution section of Introduction, how 'unactivated' and 'informative' joints are determined and to what extent?
> >
> > **R:** Thank you for this question. As mentioned in the introduction, we define unactivated joints as the ones that receive the lowest attentions scores from the MHSA blocks. We define a threshold under which we consider a joint to be unactivated, and we study the effect of this threshold in Figure  5 in the ablation studies. As for the word 'informative,' it refers to joints that could be used to identify the actions due to its unique individual and holistic movement over time. With that in mind, some unactivated could be informative. This is validated by the observation that using such joints does improve the classification performance. Please refer to Figure 1 and Table 2 which respectively provide visualizations of the having more activated joints and the qualitative results showing improved performance.

---

### Official Review · Reviewer_jVKN · 2023-10-31

**Soundness:** 2 fair
**Presentation:** 2 fair
**Contribution:** 3 good
**Rating:** 6
**Confidence:** 4

**Summary:**

This manuscript introduces MaskCLR, a training paradigm tailored for skeleton-based action recognition. This approach applies targeted masking to intentionally occlude predominantly activated joints in skeletal sequences. The authors claim that such a strategy facilitates a more comprehensive extraction of pertinent information from input skeleton joints. Complementing this, a multi-tiered contrastive learning architecture is articulated, contrasting skeleton representations at individual sample and class stratifications, culminating in a class-dissociated feature space. The implications of such a design are posited to be enhancements in model accuracy, resilience to noise perturbations, and adaptability across disparate pose estimators. Empirical results suggest that the MaskCLR paradigm exhibits superior performance metrics relative to extant methodologies on specified benchmarks.

**Strengths:**

The introduction of MaskCLR offers a distinct approach in skeleton-based action recognition. This work differentiates itself by emphasizing targeted masking, as opposed to the random masking strategies referenced in prior research by Zhu et al., 2023 and Lin et al., 2023. Ablation studies on the NTU60-XSub dataset support the claims, with an in-depth exploration of hyperparameters and a comprehensive evaluation of both Lsc and Lcc losses under varying masking strategies. The approach adopted by MaskCLR, which encodes detailed representations from input skeleton joints, demonstrates robustness against perturbations, contributing to advancements in skeleton-based action recognition.

**Weaknesses:**

The paper's comparative analysis with existing random masking methodologies seems limited in depth, particularly when referencing works like Zhu et al., 2023 and Lin et al., 2023; A more detailed juxtaposition detailing operational and performance nuances would be insightful. Another constraining factor is the exclusive reliance on attention weights from the final ST blocks, with Chefer et al., 2021 suggesting potential benefits from multi-layer attention scores. Furthermore, while the paper claims robustness and generalization, expanded empirical validation across diverse datasets might solidify these assertions. The current exploration of hyperparameters, especially the effects of Lsc weight α and Lcc weight β, could be augmented by examining their combined effects. A deeper theoretical exposition on choices made, especially regarding contrastive losses, could enhance comprehension. Lastly, a rigorous analysis detailing the model's performance under varying noise conditions would strengthen claims about its robustness against noise.

**Questions:**

1.The paper mentions a "class-dissociated feature space" as a result of the multi-level contrastive learning framework. Could the authors delve deeper into the tangible benefits of this feature space in comparison to more traditional feature spaces, especially in terms of model interpretability and robustness?
2.Given the paper's focus on robustness to perturbations, could the authors provide insights into specific types of perturbations where the model excels and where it might face challenges?
3.The robustness of MaskCLR, especially against varying degrees of noise, is paramount for real-world applicability. How does the model's performance degrade or vary with incremental noise levels in the input data?

**Details Of Ethics Concerns:**

I have no Ethics Concerns

---

> ### Author Response · Authors · 2023-11-20
> **Response to reviewer jVKN (part 1/2)**
>
> Thank you for taking the time to read our paper carefully, and for providing valuable feedback. We deeply appreciate your time and efforts, which will greatly contribute to improving this paper.
>
> > **P1:** The paper's comparative analysis with existing random masking methodologies seems limited in depth, particularly when referencing works like Zhu et al., 2023 and Lin et al., 2023; A more detailed juxtaposition detailing operational and performance nuances would be insightful
>
> **R:** We are not sure what you mean in this point. It would be greatly appreciated if you can clarify specific analysis that can contribute to evaluating the masking strategies. We do not only compare against random masking. In the appendix section A4 we study the effect on performance vs the targeted masking of different numbers of joints. The results confirm the validity of our approach in diluting the focus on a small number of joints, exploiting the information carried in other joints.
>
> > **P2:** Another constraining factor is the exclusive reliance on attention weights from the final ST blocks, with Chefer et al., 2021 suggesting potential benefits from multi-layer attention scores.
>
> **R:** Indeed, we mention this aspect as part of the limitations section of the paper. Certainly, more sophisticated saliency methods could be adopted in our framework. For simplicity, and for validating the efficacy of the approach, we experiment with taking the attention weights directly from the final MHSA block. In the Appendix section A2, we study the effect of using the attention weights from different layers and we find that the last MHSA block attention weights are indeed the best to use. We leave the more sophisticated saliency techniques to future work.
>
> > **P3:** Furthermore, while the paper claims robustness and generalization, expanded empirical validation across diverse datasets might solidify these assertions.
>
> **R:** We appreciate your feedback on this point. We want to clarify that we conduct robustness and generalization experiments under three different cases: 1) perturbed skeletons 2) skeletons from different pose estimator between training and testing and 3) perturbed skeletons from different pose estimators. In each case, we report the results under the most popular action recognition datasets, NTU60 and NTU120, under varying degrees of noise, masking, and pose estimator quality levels. Our approach consistently improves the robustness of the backbone model and skeleton perturbations as well as generalization to different pose estimators. We provide even more experiments in the appendix. Kindly let us know if there are specific datasets or different validations that would further confirm the efficacy of our approach.
>
>
> > **P4:** The current exploration of hyperparameters, especially the effects of Lsc weight α and Lcc weight β, could be augmented by examining their combined effects.
>
> **R:** Thank you for raising this point. The current model is tested with different values for α and β while applying the two losses separately. The search for better hyperparameters could indeed be improved by combining the losses under different weights and other parameters. We will conduct a more exhaustive search in our future work.
>
> > **P5:** A deeper theoretical exposition on choices made, especially regarding contrastive losses, could enhance comprehension.
>
> **R:** Thank you for pointing this out. Our proposed contrastive learning approach is an extension on the previous works on this field, especially on the self-supervised methods that are described in related works section. Due to the space limitation, we cannot delve deep into the details of contrastive learning. However, given your feedback, we will consider extending our work with more discussion on the theoretical background as part of the appendix of our paper.
>
> > **P6:** Lastly, a rigorous analysis detailing the model's performance under varying noise conditions would strengthen claims about its robustness against noise.
>
> **R:** We test our model under different types and degrees of noise from different pose estimators. As shown in section 4.3, our experiments include spatial Gaussian noise, spatiotemporal Gaussian noise, joint masking, part masking, and frame masking. Further, in the appendix section A4, we further evaluate our approach on targeted masking and shifted joints. The results empirically verify the efficacy of our method in improving robustness against different types of noise. We are open to providing further rigorous analysis under varying degrees of methods. We would greatly appreciate it if you point our specific studies or analysis that would further strengthen this work.

---

> ### Author Response · Authors · 2023-11-20
> **Response to reviewer jVKN (part 2/2)**
>
> > **P7:** The paper mentions a "class-dissociated feature space" as a result of the multi-level contrastive learning framework. Could the authors delve deeper into the tangible benefits of this feature space in comparison to more traditional feature spaces, especially in terms of model interpretability and robustness?
>
> **R:** Thank you very much, indeed that’s a very good question. We summarize the benefits of a class-dissociated feature space as follows: 1) a better-clustered feature space contributes to having a better decision boundary between different classes. Hence, this leads to faster convergence by the classification head and better classification performance in terms of top-1 accuracy. 2) Improved Model Interpretability: The "class-dissociated feature space" resulting from the multi-level contrastive learning framework signifies a representation where features are disentangled across different classes. This disentanglement can lead to enhanced model interpretability as the learned features may be more semantically meaningful and class-specific. Traditional feature spaces may lack such disentanglement, making it challenging to attribute specific features to particular classes. 3) Enhanced Robustness: The disentanglement of features in the class-dissociated space can contribute to increased robustness. Traditional feature spaces might encode information in a more intertwined manner, making models susceptible to perturbations or noise in the input data. In contrast, a class-dissociated feature space may offer a more resilient representation, where changes in one class do not unduly affect the features relevant to other classes. This characteristic can improve the model's robustness, especially when dealing with noisy or ambiguous data.
>
> > **P8:** Given the paper's focus on robustness to perturbations, could the authors provide insights into specific types of perturbations where the model excels and where it might face challenges?
>
> **R:** According to our experiments, our model performs well under spatial noise (Fig. 3), spatiotemporal noise (Fig. 3), joint masking (Fig. 4), part masking (Fig. 4), Targeted masking (Fig. 9), and shifted joints (Fig. 10). Semi-optimal performance is achieved when evaluating under less than 30% of last frames from the input skeleton sequence.
>
> > **P9:** 3.The robustness of MaskCLR, especially against varying degrees of noise, is paramount for real-world applicability. How does the model's performance degrade or vary with incremental noise levels in the input data?
>
> **R:** As shown in Figures 3, 4, 9, and 10, we evaluate MaskCLR as well as previous SOTA methods under varying degrees of noise. While the performance of all methods rapidly drops with more noise, MaskCLR shows the lowest drop in accuracy. We extend these evaluations with perturbing skeletons from different pose estimators, on which MaskCLR consistently outperforms the previous methods (see Table 4)